# Model-based prediction of spatial gene expression via generative linear mapping

Yasushi Okochi [1,2,8], Shunta Sakaguchi[3,8], Ken Nakae[4], Takefumi Kondo [3,5] & Honda Naoki[1,6,7 ✉]

Decoding spatial transcriptomes from single-cell RNA sequencing (scRNA-seq) data has become a fundamental technique for understanding multicellular systems; however, existing computational methods lack both accuracy and biological interpretability due to their model-free frameworks. Here, we introduce Perler, a model-based method to integrate scRNA-seq data with reference in situ hybridization (ISH) data. To calibrate differences between these datasets, we develop a biologically interpretable model that uses generative linear mapping based on a Gaussian mixture model using the Expectation–Maximization algorithm. Perler accurately predicts the spatial gene expression of *Drosophila* embryos, zebrafish embryos, mammalian liver, and mouse visual cortex from scRNA-seq data. Furthermore, the reconstructed transcriptomes do not over-fit the ISH data and preserved the timing information of the scRNA-seq data. These results demonstrate the generalizability of Perler for dataset integration, thereby providing a biologically interpretable framework for accurate reconstruction of spatial transcriptomes in any multicellular system.

[1] Laboratory for Theoretical Biology, Graduate School of Biostudies, Kyoto University, Kyoto, Japan. [2] Faculty of Medicine, Kyoto University, Kyoto, Japan. [3] Laboratory for Cell Recognition and Pattern Formation, Graduate School of Biostudies, Kyoto University, Kyoto, Japan. [4] Graduate School of Informatics, Kyoto Universityo, Kyoto, Japan. [5] The Keihanshin Consortium for Fostering the Next Generation of Global Leaders in Research (K-CONNEX), Kyoto, Japan. [6] Laboratory for Data-driven Biology, Graduate School of Integrated Sciences for Life, Hiroshima University, Higashi-Hiroshima, Hiroshima, Japan. [7] Theoretical Biology Research Group, Exploratory Research Center on Life and Living Systems (ExCELLS), National Institutes of Natural Sciences, Okazaki, Aichi, Japan. [8] These authors contributed equally: Yasushi Okochi, Shunta Sakaguchi. ✉email: nhonda@hiroshima-u.ac.jp

Genes are heterogeneously expressed in multicellular systems, and their spatial profiles are tightly linked to biological functions. In developing embryos, spatial gene-expression patterns are responsible for coordinated cell behavior (e.g., differentiation and deformation) that regulates morphogenesis[1]. In addition, within organ tissues, cells at different locations play different roles in organ function based on their gene-expression patterns[2]. Thus, identification of spatial genome-wide gene-expression profiles is key to understanding the functions of various multicellular systems. In situ hybridization (ISH) has been widely used to visualize spatial profiles of gene expression; however, application of this method is generally limited to only small numbers of genes. By contrast, the single-cell RNA-sequencing (scRNA-seq) method developed during the previous decade has enabled measurement of genome-wide gene-expression profiles in tissues at the single-cell level[3]. However, this method requires tissue dissociation, which leads to loss of spatial information for the original cells.

To compensate for the lost spatial information, new computational approaches have emerged (Seurat (v.1)[4], DistMap[5], Achim et al.[6], Halpern et al.[7]), enabling reconstruction of genome-wide spatial expression profiles from scRNA-seq data by integrating existing ISH data as a spatial reference map in silico. However, their methods require binarization of gene-expression data[8], which leads to unsatisfactory accuracy, or tissue-specific modeling, which leads difficulty in application to other systems. Recently, the seminal methods Seurat (v.3)[9] and Liger[10,11] were developed to address gene-expression data as continuous variables in a non-tissue-specific manner. These methods match the distributions of ISH and scRNA-seq data points by using dimensionality reductions [e.g., canonical correlation analysis (CCA)][12] and integrative nonnegative matrix factorization (iNMF)[13], followed by mapping the scRNA-seq data points to the nearest ISH data points, according to Euclidean distance using nearest-neighbor (NN) methods (e.g., $k$-NN[14] and mutual NN[15]). However, a major issue is that the flexibility of the methods allow mapping of ISH data to scRNA-seq data without any models of the underlying scRNA-seq data structure. Specifically, these methods do not account for difference in gene-expression noise associated with each gene. Given this model-free property, these methods are dependent upon nonlinear NN mapping, which innately causes overfitting to the reference ISH data.

To address these issues, we propose a model-based computational method for probabilistic embryo reconstruction by linear evaluation of scRNA-seq (Perler), which reconstructs spatial gene-expression profiles via generative linear modeling in a biologically interpretable framework. Perler addresses gene-expression profiles as continuous variables and models generative linear mapping from ISH data points into the scRNA-seq space. To estimate parameters of the linear mapping, we developed a method based on the expectation–maximization (EM) algorithm[14]. Using the estimated parameters, we also propose an optimization method to infer spatial information of scRNA-seq data within a tissue sample. We applied this method to existing *Drosophila* scRNA-seq data[5] and successfully reconstructed spatial gene-expression profiles in *Drosophila* early embryos that were more accurate than those generated using another spatial reconstruction method (DistMap[5]). In addition, we showed that Perler can reconstruct a spatial gene-expression pattern that could not be fully predicted using previous methods, including Seurat (v.3), Liger, and DistMap. Further analysis revealed that Perler was able to preserve the timing information of the scRNA-seq data without overfitting to the reference ISH data. Furthermore, we demonstrated that this method accurately predicted spatial gene-expression profiles in early zebrafish embryos[4], the mammalian liver[7], and the mouse visual cortex[16,17]. These findings demonstrate Perler as a robust, generalized framework for predicting spatial transcriptomes from any type of ISH data for any multicellular system without overfitting to the reference.

## Results

**Framework of spatial reconstruction in Perler**. Perler is a computational method for model-based prediction of spatial genome-wide expression profiles from scRNA-seq data that works by referencing spatial gene-expression profiles measured by ISH (Fig. 1a). In general, scRNA-seq data have higher dimensionality (on the order of ~10,000 genes), but does not contain information of spatial coordinates in tissues. By contrast, reference ISH data contain expression information for $D$ genes in each cell or tissue subregion, with these referred to as landmark genes (e.g., $D = 84$ in *Drosophila melanogaster* early embryos) and tagged with spatial coordinates in tissues.

The Perler procedure involves two steps. The first step estimates a generative linear model-based mapping function that transforms ISH data into the scRNA-seq space, thereby enabling calculation of pairwise distances between ISH data and scRNA-seq data (Fig. 1b). The second step reconstructs spatial gene-expression profiles according to the weighted mean of scRNA-seq data, which is optimized by the mapping function estimated in the first step (Fig. 1c).

The first step considers gene-specific differences between scRNA-seq and ISH measurements. For example, we assume that some genes are more or less sensitive to ISH or scRNA-seq, and subject to high or low background signals in the associated data. We account for gene-specific noise intensity, because gene expression fluctuates over time in a gene-specific manner[18]. These differences in sensitivity, background signals, and noise intensity can be expressed by linear mapping:

$$y_i = a_i h_i + b_i + c_i \xi_i$$

where $y_i$ and $h_i$ denote the expression levels of landmark gene $i$ measured by scRNA-seq and ISH, respectively; $a_i$, $b_i$, and $c_i$ are constant parameters of gene $i$ and interpreted as the sensitivity coefficient, background signal, and noise intensity, respectively; and $\xi_i$ indicates standard Gaussian noise. Note that $a_i$, $b_i$, and $c_i$ are different for each gene, and that these parameter values are unknown. To estimate this linear mapping from the data, we developed a generative model in which scRNA-seq data points are generated/derived from each cell in the tissue, whose expression is measured by ISH (see "Methods"). We then derived a parameter estimation procedure based on the EM algorithm (see "Methods"). Using the estimated parameters, a gene-expression vector for each cell in a given tissue sample measured by ISH can be mapped to the scRNA-seq space, thereby allowing evaluation of pairwise distances between ISH and scRNA-seq data.

The second step reconstructs the spatial gene-expression profile in tissue from scRNA-seq data. We estimated gene expression of each cell in a tissue sample according to the weighted mean of all scRNA-seq data points, where the weights were determined by the pairwise distances between cells in tissue samples measured using ISH and scRNA-seq data points (Fig. 1c). Weights were evaluated by Mahalanobis distance, which accounts for the reliability of each gene depending on its noise intensity (Fig. 1d, Eqs. (22–25)). For the best prediction, we optimized the hyperparameters of the weighting function to ensure that the predicted and referenced landmark gene-expression profiles were well-correlated by cross-validation (CV; Fig. 1e and see "Methods"). We then predicted non-landmark gene expression using the optimized weighting function. Data were preprocessed before the first step. Some landmark genes redundantly exhibit similar spatial expression

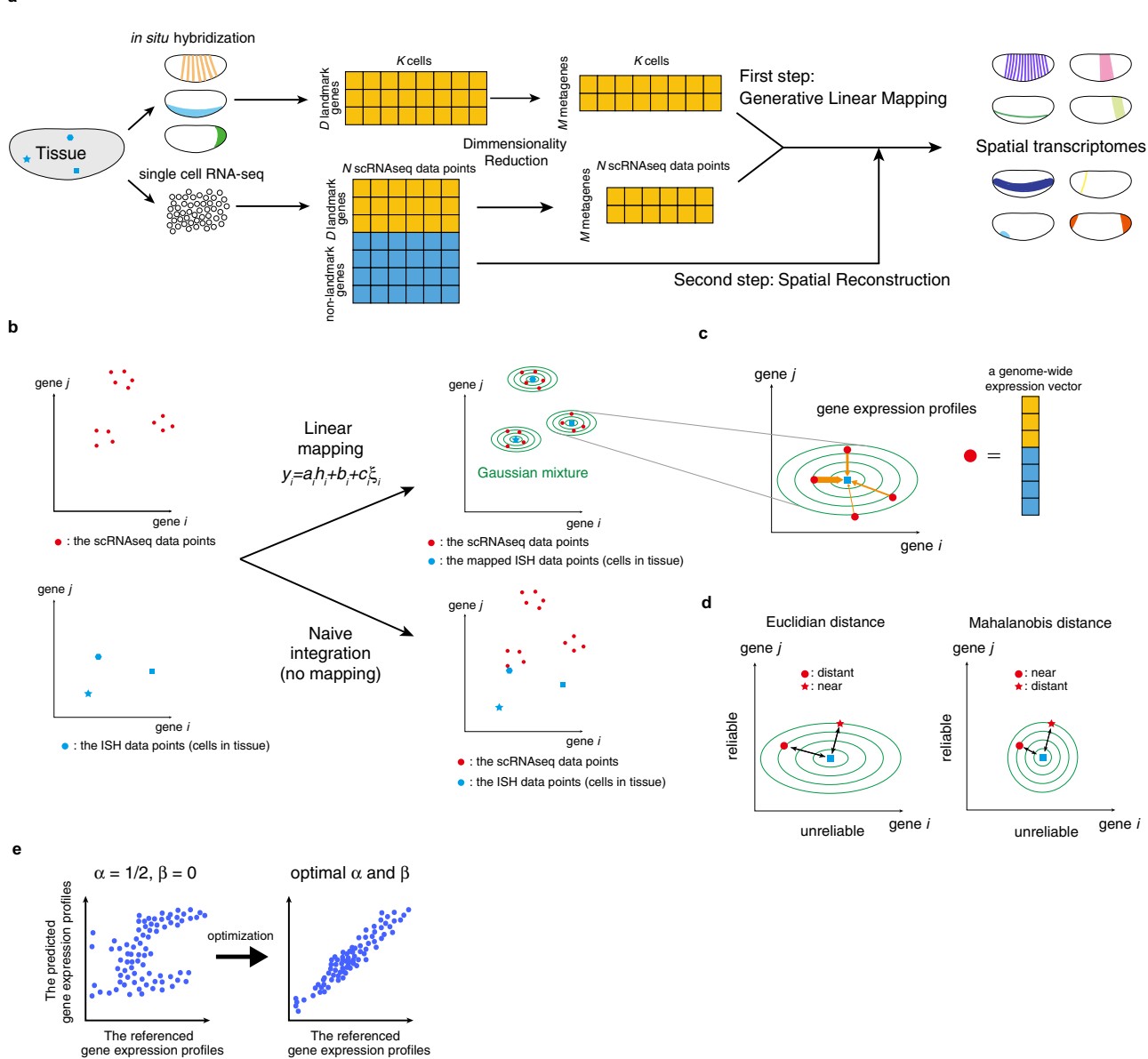

**Fig. 1 Schematic illustration of Perler. a** Flow of data processing. **b** Generative linear mapping from ISH data to the scRNA-seq space. The left and right panels indicate scatter plots in high-dimensional ISH and scRNA-seq spaces. Because ISH data points did not match scRNA-seq data points (naive integration) in the absence of mapping, ISH data points were mapped in order to fit the scRNA-seq data points the best using the EM algorithm. Blue points indicate ISH data points, red points indicate scRNA-seq data points, and green lines indicate contours of the estimated multivariate Gaussian distribution (see "Methods"). **c** Reconstruction/prediction of gene expression by Mahalanobis' metric-based weighting (see "Methods"). Orange arrows indicate weights between scRNA-seq data points to cell $k$, and their widths reflect the Mahalanobis' metric-based weights. **d** Schematic demonstrating the difference between Euclidian and Mahalanobis distance. The expression levels of gene $i$ are not reliable as this gene has high noise intensity, while gene $j$ has low noise intensity, thus, its expression levels are considered reliable. In the Mahalanobis distance, the expression levels of each gene are considered by a variance scale. Note, although the "star" scRNA-seq data point is nearer to the ISH data point than the "circle" scRNA-seq data point in Euclidian distance, the weight of the "star" data point is smaller than that of the "circle" point. **e** Weight determination. The hyperparameters of the weighting function, $\alpha$ and $\beta$, are determined by cross-validation to ensure that the referenced gene-expression profiles correlate well with the predicted gene-expression profiles (see "Methods"). Dots correspond to cells in tissue. The left and right panels indicate the conceptual scatter plots of the expression levels of the genes before ($\alpha = 1/2$, $\beta = 0$) and after parameter optimization, respectively.

patterns, which can lead to biased parameter estimation and cause a loss of mapping ability. To reduce redundancy in scRNA-seq and ISH data, we performed dimensionality reduction using partial least squares correlation analysis (PLSC)[19] (see "Methods"). Each factor in the reduced dimension can be interpreted as a "metagene", which is representative among a highly correlated gene cluster, with its coordinate corresponding to the expression level of the metagene. In Perler, we regarded

the metagene-expression level $i$ (i.e., factor $i$) in the scRNA-seq and ISH spaces as $y_i$ and $h_i$ in the equation above (see "Methods").

**Model-based mapping between scRNA-seq and ISH data.** Previously, Karaiskos et al.[5] measured gene expression in individual cells dissociated from early *D. melanogaster* embryos at

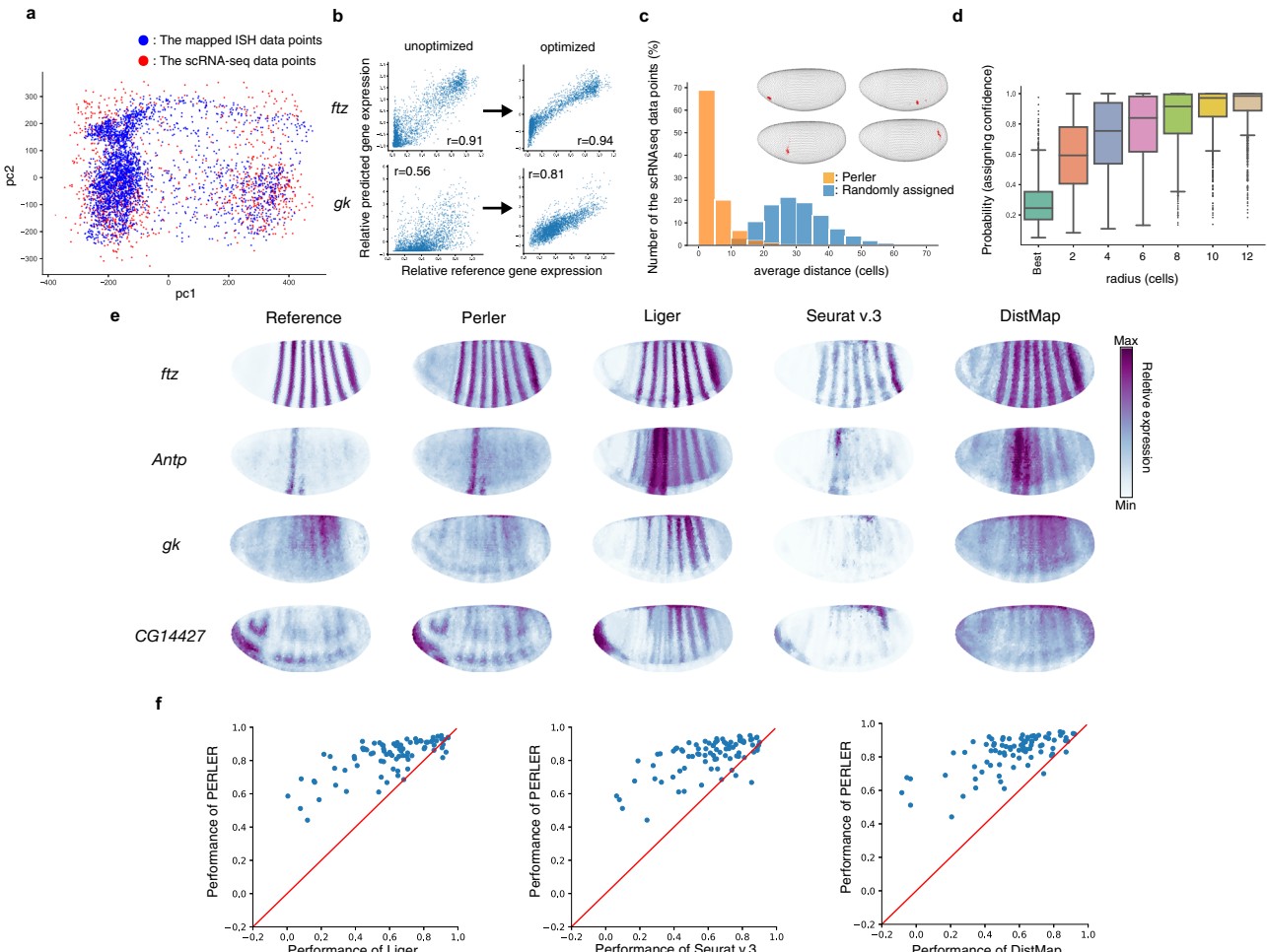

**Fig. 2 Generation of spatial gene-expression profiles. a** Scatter plot of mapped gene expression and scRNA-seq observations (Fig. 1b, upper right panel). Principal component analysis[13] was used to visualize high-dimensional gene-expression data into two dimensions. **b** Improved correlation between predicted and referenced data in the scRNA-seq space by optimizing the weighting function. **c** Histogram of the assigned specificity evaluated by the distance between the best assigned location and the following best three locations. The distance was calculated by mean path length on the $k$-NN graph comprising all cells in the tissue ($k = 6$). Examples of estimated original positions of each scRNA-seq data point (inset). Embryos are colored according to the posterior probabilities for the scRNA-seq data points. Point sizes indicate the magnitude of each posterior probability. Points with posterior probabilities <0.001 were omitted. **d** Boxplot of the assigned specificity calculated as the posterior probabilities of circular regions for each scRNA-seq data point according to radius, with the center of each region representing the best assigned location for each data point. For the box signifies the upper and lower quartiles, and the median is represented by a short black line within the box. The boxplot has whiskers with a maximum 1.5 interquartile range, with black points indicating outliers. The radius was calculated by path length on the $k$-NN graph comprising all cells in the tissue. $n = 1297$ biologically independent cells (scRNA-seq data points). **e** Reconstructions of the landmark genes by Perler, Liger, Seurat (v.3), and DistMap. **f** Comparison of Perler reconstruction performance with Liger (left, two-sided Wilcoxon test: $p = 1.8 \times 10^{-14}$), Seurat (v.3) (middle, two-sided Wilcoxon test: $p = 2.5 \times 10^{-14}$), and DistMap (right, two-sided Wilcoxon test: $p = 2.9 \times 10^{-15}$). Each dot indicates the reconstruction accuracies for each gene by Perler and other methods. Red lines depict auxiliary lines showing the same performance of two methods. Source data are provided in a Source data file.

developmental stage 6 by scRNA-seq, followed by development of a computational method (DistMap) to reconstruct the spatial gene-expression profile of the embryos from the scRNA-seq data. They used as reference data a spatial gene-expression atlas provided by the Berkeley *Drosophila* Transcription Network Project (BDTNP)[20,21], in which the expression of 84 landmark genes was quantitatively measured by fluorescent (FISH) at single-cell resolution at developmental stage 5.

In the present study, we applied Perler to the same scRNA-seq dataset and used the 84 landmark genes from the BDTNP atlas as the spatial reference map. We then predicted the spatial gene-expression profiles for 8840 non-landmark genes. To compare Perler results with those of DistMap, we used the same normalization methods for the scRNA-seq dataset as the previous study[5]. For preprocessing, we manually extracted 60 metagenes as

nonredundant clusters of the landmark genes by dimensionality reduction, because some landmark genes were correlatedly expressed in both the ISH and the scRNA-seq data (Supplementary Fig. 1a, b). Then, Perler estimated the parameters of the linear mapping by integrating the scRNA-seq data with the ISH data (Supplementary Fig. 2a–c). The mapped ISH data points according to the linear mapping were distributed consistently with the scRNA-seq data points (Fig. 2a and Supplementary Fig. 2d). We also confirmed that the linear mapping properly calibrated the difference between ISH and scRNA-seq data on each metagene level (Supplementary Fig. 3).

Perler can predict the origin of a scRNA-seq data point in a tissue by computing a posterior probability that a scRNA-seq data point was generated from each cell in the tissue sample. We found that the scRNA-seq data points were specifically assigned to cells

in a small region (a few cell diameters) of the tissue (Fig. 2c, d and Supplementary Fig. 2e). We also evaluated this performance of other methods (Supplementary Fig. 4), showing that Perler had superior performance to DistMap and equivalent performance to Liger and Seurat v.3.

Based on linear mapping, Perler reconstructed gene-expression profiles from scRNA-seq data. The reconstructed and referenced gene-expression profiles were well-correlated following optimization of the hyperparameters (Fig. 2b and Supplementary Fig. 5). The reconstruction accuracy of Perler (average correlation coefficient (aCC) = 0.83) was significantly higher than that of Seurat v.3 (aCC = 0.61), Liger (aCC = 0.61), and DistMap (aCC = 0.56; Fig. 2e, f and Supplementary Fig. 6).

We also evaluated the predictive performance of Perler by conducting leave-one-gene-out cross-validation (LOOCV) to confirm whether gene expression can be predicted following removal of the landmark gene of interest from the ISH data prior to training (Supplementary Figs. 7 and 8). The predictive accuracy of Perler (aCC = 0.59) was significantly higher than that of Seurat v.3 (aCC = 0.55), Liger (aCC = 0.51), and DistMap (aCC = 0.44; Supplementary Fig. 7). However, we noticed that some genes lost the predictive accuracy compared with the reconstruction accuracy (Supplementary Fig. 9a), indicating that genes can be classified as well-predicted or poorly predicted. To clarify the difference between these two classes of genes (Supplementary Table 1), we examined the correlated data structure among landmark genes (Supplementary Fig. 9b–d) and found that poorly predicted genes had different expression patterns between ISH and scRNA-seq (Supplementary Fig. 9d), suggesting that the loss of prediction accuracy was primarily caused by different correlated data structures between ISH and scRNA-seq. Further, we showed that the predictive accuracy of each well-predicted gene was not affected by eliminating the poorly predicted genes (aCC = 0.64 before and 0.63 after gene elimination), indicating that Perler robustly predicted the spatial gene-expression pattern irrespective of the poorly predicted genes.

These results demonstrated that Perler accurately reconstructed and predicted the spatial expression profiles of the landmark genes and was capable of doing this via simple linear mapping.

**Validity and robustness of Perler**. Next, to analyze the validity of Perler, we examined the effect of dimensionality reduction on reconstruction and predictive accuracy. We found that the high reconstruction accuracy (aCC > 0.78) was maintained regardless of the presence of the dimensionality reduction, whereas the predictive accuracy was significantly improved by dimensionality reduction (Supplementary Fig. 10a, b), indicating that dimensionality reduction was an important factor for avoiding overfitting to ISH data. We also examined the effect of optimizing hyperparameters and found that it significantly improved Perler performance (Supplementary Fig. 11a).

Moreover, we analyzed the robustness of Perler against the downsampled landmark gene set by, first, evaluating the reconstruction performance under random selection of different quantities of landmark genes (Supplementary Fig. 12a). We found that reconstruction accuracy increased with the number of landmark genes, while only 30 landmark genes were required to achieve comparable performance with Liger and Seurat v.3, with full use of the landmark genes (>0.6). Second, we evaluated the resolution of the predicted origin of a scRNA-seq data point (Supplementary Fig. 13a), and found that the confidence for origin prediction of a scRNA-seq data point to a small region increased with the number of landmark genes, while only 40 landmark genes were required for a scRNA-seq data point to

be predicted to a small region (four-cell radius) with sufficient confidence (>0.5). To further demonstrate the robustness of Perler, we conducted tenfold CV, for which the folds were extracted from the *Drosophila* scRNA-seq data points[22]. In this CV scheme, the performance of methods was evaluated using scoring metrics that were previously used in the DREAM Single-Cell Transcriptomics challenge[22]. We found that Perler more robustly performed the origin prediction of the scRNA-seq data points compared with other methods (Liger and Seurat v.3), although the top-ranked methods developed in the DREAM challenge had better performance than Perler (Supplementary Tables 2–4). Note that the metrics used in this DREAM challenge were designed assuming that DistMap prediction was ground truth, namely, high scores on these metrics would be interpreted as a method that performs similarly to DistMap.

Taken together, we conclude that, in terms of reconstructing gene-expression profiles and predicting the original location of scRNA-seq data points, Perler exhibits robustness against the downsampled gene set and the tenfold CV of scRNA-seq data point.

**Prediction of non-landmark genes**. In addition to the landmark genes, Perler successfully predicted the spatial expression profiles of non-landmark genes along both anterior–posterior (A–P) and dorsoventral (D–V) axes (Fig. 3a, b). Furthermore, we evaluated the predicted spatial profile of 310 spatially restricted genes proposed by Bageritz et al.[23] (Supplementary Figs. 14 and 15) and found that Perler was able to uncover the unknown spatial gene-expression pattern. Notably, we observed that spatial patterns predicted by Seurat (v.3), Liger, and DistMap were incomplete. For example, the predicted stripes disappeared in the ventral part of embryos (e.g., *abd-A* and *Ubx* in Fig. 3a), whereas this issue was not observed with Perler, which accurately predicted the stripe pattern, even in the ventral part of embryos.

**Prediction of 14-stripe patterns of segment-polarity genes**. We then presented the spatial predictions of "segment-polarity" genes, which are expressed in a 14-stripe pattern consistent with the parasegments that subdivide the trunk (main body) region of embryos (Fig. 3b)[24–28]. Although the BDTNP reference does not contain information concerning the genes expressed in the 14-stripe pattern, we found that Perler accurately predicted the spatial expression patterns of these segment-polarity genes, including *engrailed* (*en*), *wingless* (*wg*), *hedgehog* (*hh*), and *midline* (*mid*) (Fig. 3b)[24–28]. By contrast, all of the previous methods exhibited issues regarding prediction of the 14-stripe patterns. The predicted patterns demonstrated that DistMap and Seurat (v.3) were unable to predict any 14-stripe patterns, and that Liger partially predicted 14-stripe patterns, although the ventral part of each stripe was missing (Fig. 3b). These results suggested that Perler more accurately revealed the spatial gene-expression patterns of non-landmark genes.

We further analyzed the details of the gene-expression profiles of the segment-polarity genes within each parasegment. Each parasegment shows a four-cell width and is delimited by periodic expression of pair-rule genes and segment-polarity genes at the single-cell width resolution at stage 6[29,30] (Fig. 4a). First, we confirmed that the reconstructed patterns of *ftz*, *eve*, and *odd* were consistent with experimental results (Fig. 4b). In addition, the predicted stripes of *wg* were identified adjacent to the predicted stripes of *en*, and the predicted stripes of *en* were identified adjacent to the reconstructed stripes of *odd* (Fig. 4b, c). These results were consistent with experimental results[29,30],

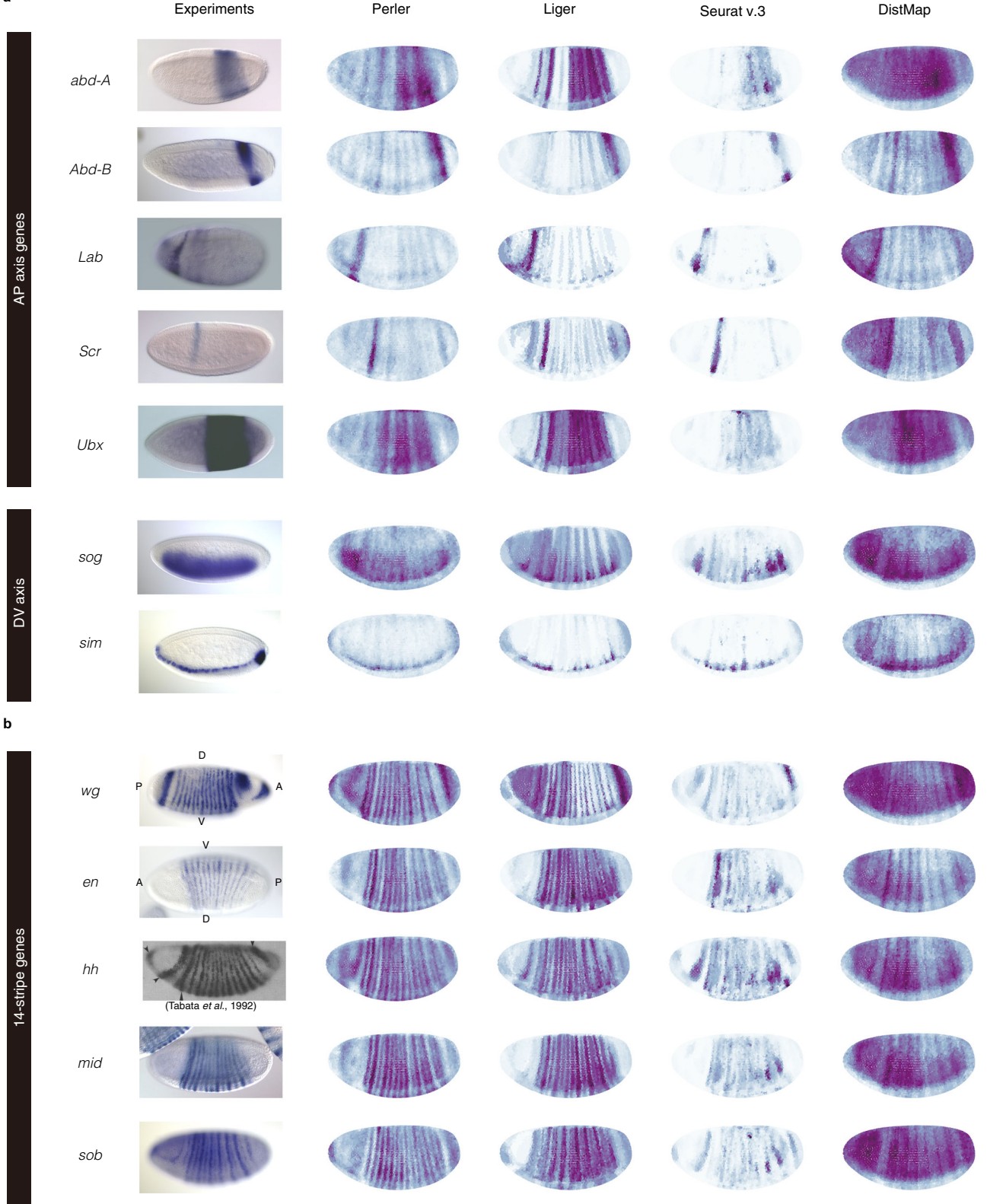

**Fig. 3 Spatial prediction of non-landmark genes.** Predictions of non-landmark gene expression showing **a** spatial expression along the A–P and D–V axes and **b** a 14-stripe pattern according to Perler, Liger, Seurat (v.3), and DistMap. ISH images (other than *hh*) were adapted from BDGP (Berkeley Drosophila Genome Project)[25] under a Creative Commons License (Creative Commons CC0 license). ISH image of *hh* was reprinted from Tabata et al.[24] under a Creative Commons License (Attribution: Non-Commercial 4.0 International License).

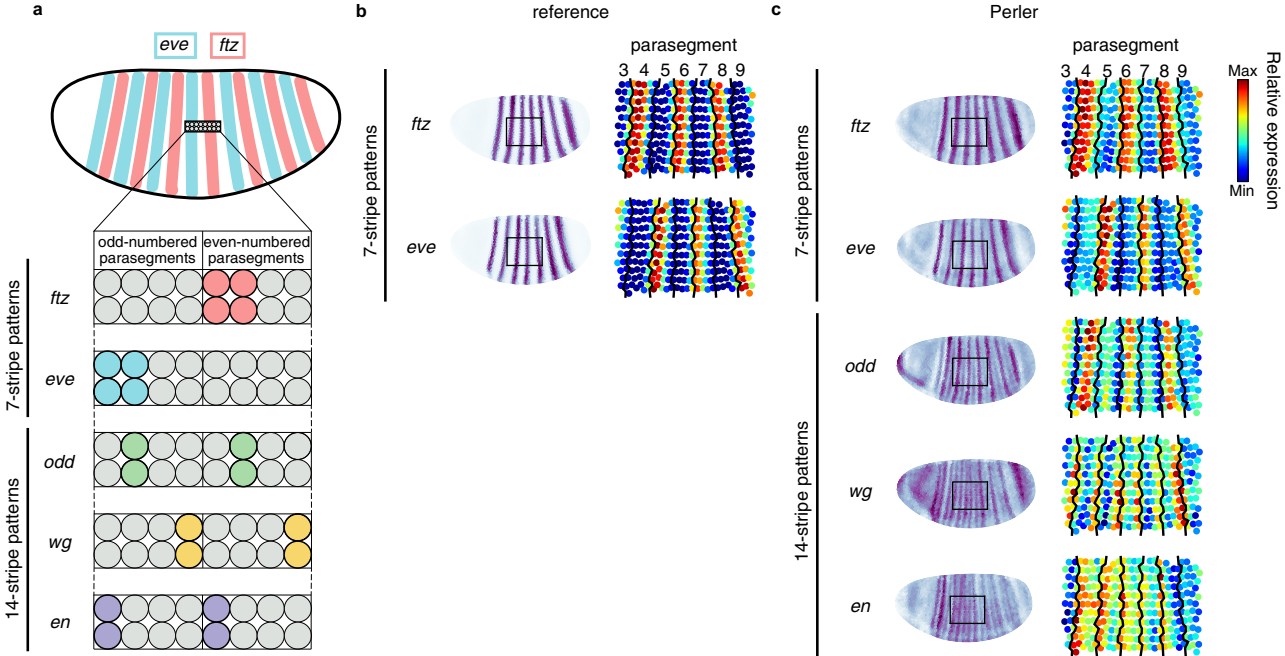

**Fig. 4 Perler prediction at single-cell resolution. a** Spatial expression profiles of pair-rule and segment-polarity genes at the single-cell level within each parasegment[29,30]. **b** The referenced stripe patterns of *ftz* and *eve*. **c** The reconstructed stripe patterns of *ftz*, *eve*, *odd*, *wg*, and *en* at single-cell resolution. The left panel shows the spatial gene-expression profiles generated by Perler. The right panel shows the expanded image of the left panel. The black square in the left panels is the region of interest in the right panels. Black lines in the right panels of **b** and **c** indicate the boundaries of each parasegment, which were determined by expression patterns of *ftz* and *eve* in the reference ISH dataset (**b**).

strongly supporting the ability of Perler to reveal differences in spatial gene expression at single-cell resolution.

**Preservation of timing information of scRNA-seq data**. We then investigated the effect of timing differences between scRNA-seq (stage 6) and FISH (stage 5) experiments. Although most gene-expression patterns at stage 6 are the same as those at stage 5, several "pair-rule" genes (*odd*, *prd*, *slp1*, and *run*) exhibit stripe-doubling from the 7- to the 14-stripe expression patterns during stages 5 and 6[29] (Fig. 5a). Accordingly, the scRNA-seq data should intrinsically contain information for the 14-stripe expression pattern. Therefore, we determined whether Perler could reconstruct the 14-stripe pattern from the stage 6 scRNA-seq data.

In our reconstruction, *ftz*, *eve*, and *h* showed a seven-stripe pattern, which was consistent with the previous report[25] showing that these genes do not exhibit stripe-doubling during stages 5 and 6 (Fig. 5b). For *odd*, *prd*, and *slp1*, which exhibit stripe-doubling, Perler reconstructions resulted in 14-stripe patterns (Fig. 5c). In addition, reconstruction of *run* resulted in a partial stripe-doubling pattern, where the third stripe from the posterior of the embryo was split into two stripes (Fig. 5c), surprisingly suggesting that Perler detected the ongoing phase of a 7-stripe to 14-stripe pattern. These results showed that Perler was able to reconstruct embryos according to the timing of the scRNA-seq experiment. By contrast, Seurat (v.3) and DistMap reconstructed every pair-rule gene as seven-stripe patterns (Fig. 5b, c). Moreover, Liger reconstructed *odd*, *prd*, and *slp1* as broad primary seven stripes with weak secondary seven stripes, which were so obscure that it was difficult to distinguish 14 stripes, and reconstructed *run* as a 7-stripe pattern (Fig. 5c). These results indicated that previous methods reconstructed embryos according to the timing of FISH experiments rather than that of scRNA-seq experiments. Taken together, these findings showed that Perler successfully reconstructed spatial gene-expression profiles according to the timing of scRNA-seq experiments (stage 6),

regardless of the timing of FISH experiments (stage 5), while all other methods reconstructed those at the timing of FISH experiments. We concluded that Perler has the ability to not over-fit to ISH data and robustly preserve timing information in scRNA-seq data.

**Application to other datasets**. To evaluate Perler applicability to other datasets, we evaluated it using three published datasets. First, we applied Perler to the zebrafish embryo datasets (Supplementary Fig. 1, 10–13, 16, and 17), in which the spatial reference map was binarized based on traditional measurement by ISH[4] (Fig. 6a and see "Methods"). LOOCV demonstrated that Perler accurately predicted spatial gene-expression profiles compatible with Seurat (v.1)[4] (median receiver operating characteristic (ROC) score = 0.97), even using the binary spatial reference map (Fig. 6a–c).

We then applied Perler to mammalian liver datasets (Supplementary Figs. 11, 18, and 19), in which the spatial reference map was measured by single-molecule (sm)FISH[7] (Fig. 6d and see "Methods"). LOOCV showed that the predictive accuracy (aCC = 0.87) was sufficiently high (Fig. 6e), and that Perler successfully predicted both monotonic and non-monotonic gene-expression gradients (Fig. 6f)[7].

Finally, we applied Perler to adult mouse visual cortex datasets (Supplementary Figs. 1, 11–13, 20, and 21), in which the single-cell resolution ISH data for 1020 genes was measured by recent in situ technology (STARmap[16]), and scRNA-seq data for 14,739 cells available from the Allen Brain Atlas[17]. CV revealed that Perler predicted the spatial expression patterns of genes according to both layer-specific expression and cell-type-specific expression in brain cortex (Fig. 6g). We also applied Perler to another scRNA-seq dataset (194,027 cells; Drop-viz[31]), and found that it predicted the spatial gene-expression patterns using the Drop-viz dataset consistently with the Allen Brain Atlas dataset (Supplementary Figs. 22–24). These results suggest that Perler

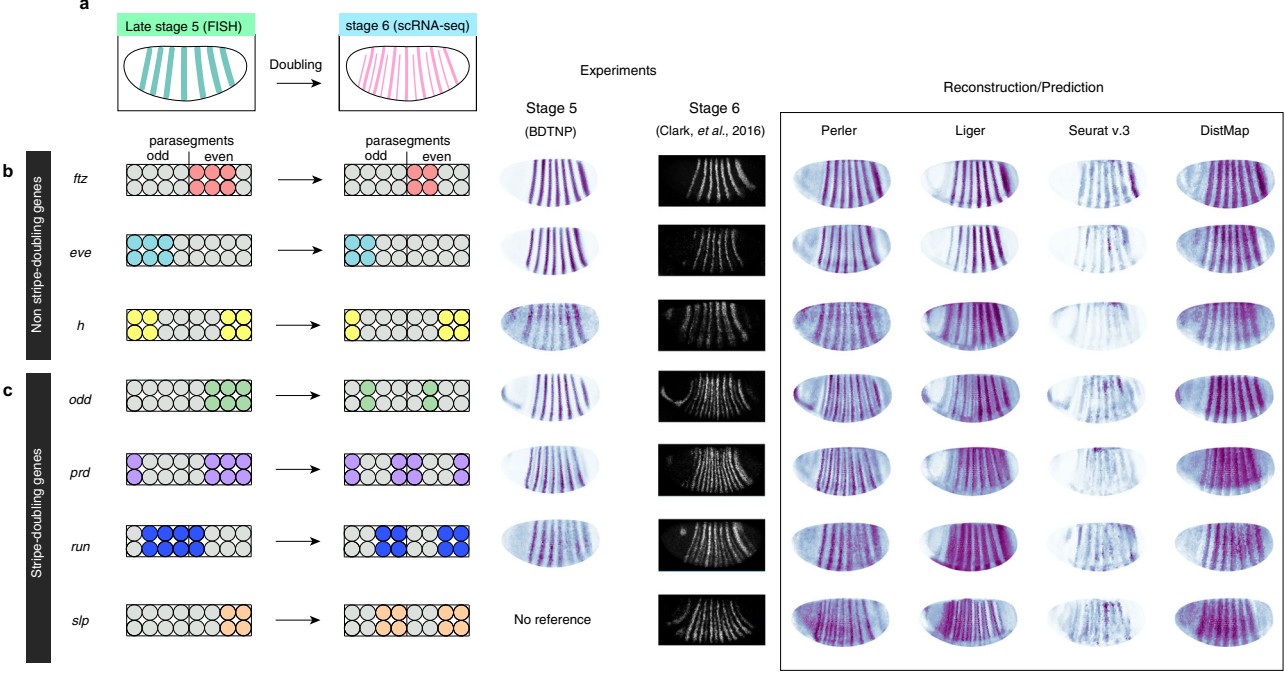

**Fig. 5 Generalization to spatial reference maps by Perler. a** Stripe-doubling of pair-rule genes from *Drosophila* developmental stage 5 (FISH experiment) to stage 6 (scRNA-seq experiment). **b, c** Left panels: images of expression changes in **b** non-stripe-doubling genes and **c** stripe-doubling genes. Middle panels: experimental ISH data of stage 5 (BDTNP) and stage 6 (Clark et al.[29]). Right panels: gene expression reconstructed/predicted by Perler, Liger, Seurat (v.3), and DistMap. Note that *slp* expression was predicted. Experimental ISH data were reprinted from Clark et al.[29] under a Creative Commons License (Attribution 4.0 International License).

is applicable for prediction using high-dimensional spatial reference maps.

Taken together, the findings support Perler as a powerful tool for predicting spatial gene-expression profiles in any multicellular system with general applicability to any type of ISH data (e.g., binary or continuous, low to high dimension, and single-cell to tissue-level resolution).

## Discussion

In this study, we developed a model-based computational method (Perler) that predicts genome-wide spatial transcriptomes. Perler sequentially conducted a two-step computation, with the first step mapping ISH data points to the scRNA-seq space according to the generative linear model by EM algorithm (Fig. 1b), and the second step optimizing the weighting function used to predict spatial transcriptomes according to weighted scRNA-seq data points (Fig. 1c, d). Using a dataset for early *Drosophila* embryos, we demonstrated that Perler accurately reconstructed and predicted genome-wide spatial transcriptomes with robustness (Figs. 2–5). Moreover, we showed that in any multicellular system, Perler displayed broad applicability to any type of ISH data (Fig. 6).

We propose that Perler offers three innovative features. First, Perler can calibrate the difference between scRNA-seq and ISH measurement properties. To express this difference, we applied a "linear mapping model" assuming biologically interpretable constraints that expression levels are linearly correlated between ISH and scRNA-seq measurements with gene-specific sensitivity, background signals, and noise intensity, as in Eq. (1). Second, Perler can reliably reconstruct gene-expression patterns in a noise-resistant manner. Specifically, Perler can evaluate to which extent each gene is reliable for reconstruction depending on the noise intensity (Fig. 1d), by using Mahalanobis pairwise distances (Eq. (23)), related to Fig. 1c). As a

result, more reliable genes with low noises have larger contribution to the weights for the reconstruction, whereas less reliable genes with high noises have smaller contribution. It should be stressed that such quantitative evaluation of gene reliability is possible only with a method using a generative model. Third, the model-based linear mapping used in Perler is beneficial in terms of the performance for gene-expression pattern reconstruction. To ensure generalized performance, we introduced generative linear modeling with biologically interpretable constraints and statistically reasonable distances. This model-based characteristic of Perler differs from Seurat (v.3)[9] and Liger[10], both based on model-free mapping between ISH and scRNA-seq data (e.g., CCA and NMF methods). Their model-free mapping addresses gene expression as continuous variables with applicability to any kind of multicellular system; however, these methods freely map ISH data to scRNA-seq data without any assumptions (i.e., they do not account for latent relationships between the two datasets). We showed that model-based Perler significantly improved reconstruction/prediction accuracy compared with other methods (Liger, Seurat v.3, and DistMap; Fig. 2e, f). Further, in a demonstration using *Drosophila* data, Perler was found to preserve the timing information of scRNA-seq data and robustly reconstruct the spatial gene-expression patterns of the pair-rule genes; whereas this kind of robustness is not observed in other model-free methods (Liger, Seurat v.3, and DistMap; Fig. 5). For example, by focusing on the stripe-doubling of pair-rule genes in *Drosophila*, Perler successfully reconstructed 14-stripe patterns at a single-cell resolution, while Seurat v.3 and Liger were unable to effectively reconstruct these patterns, and over-fit to the timing of ISH experiments (Fig. 5). We believe that these results highlight the importance of using model-based prediction of spatial gene-expression patterns. Additional characteristic features of Perler are summarized in Supplementary Table 5.

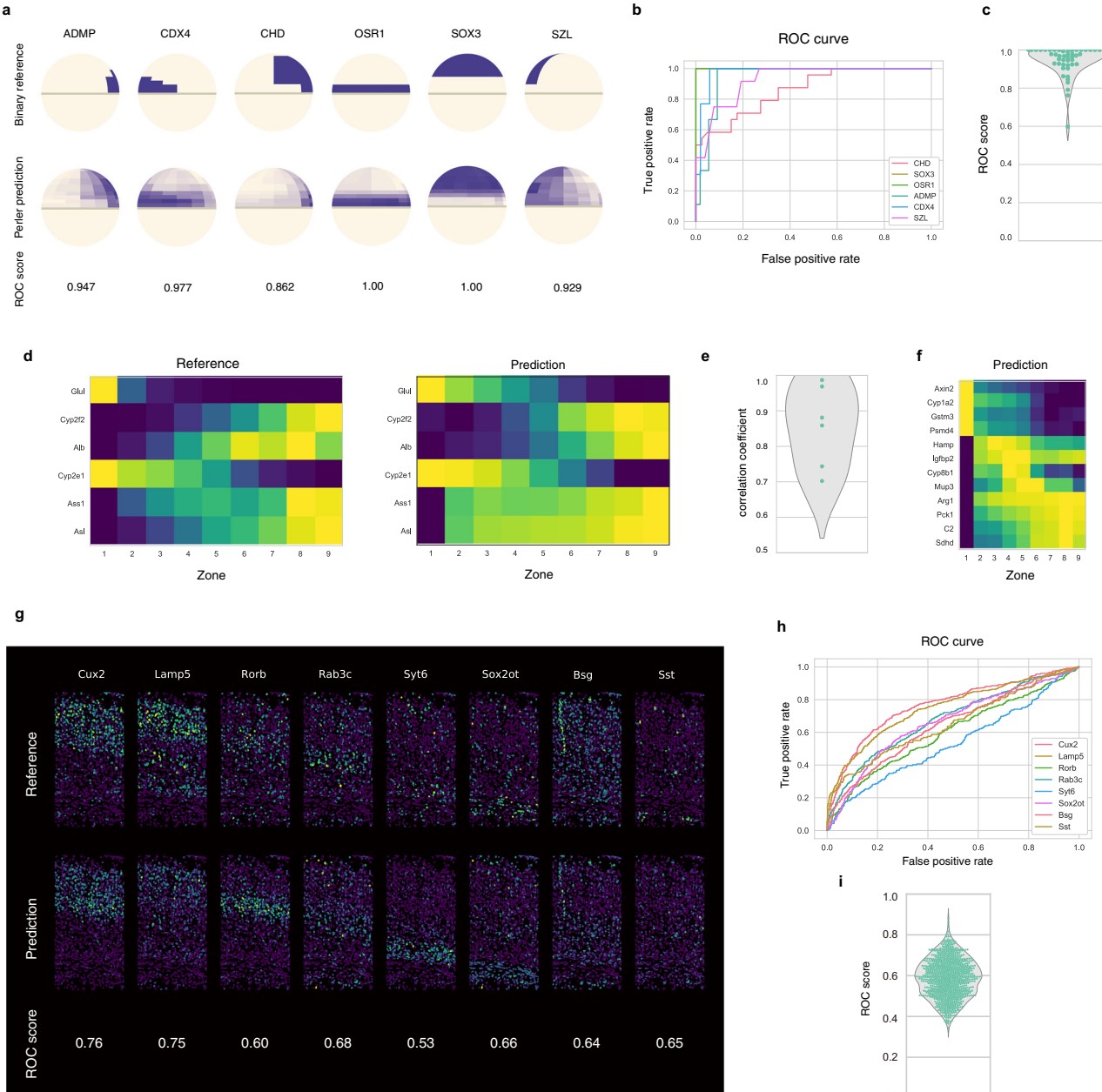

**Fig. 6 Applications of Perler to other data. a–c** Application of Perler for early zebrafish embryo data. **a** LOOCV experiments. The upper and lower panels show the referenced ISH data and predicted gene-expression profiles, respectively. **b** ROC curve for the LOOCV experiments for the genes shown in **a**. **c** Violin plot for the predictive accuracies of Perler for the LOOCV experiments for all genes in the reference ISH data according to ROC score. **d–f** Application of Perler for mammalian liver lobules. **d** LOOCV experiments. The left and right panels show the reference ISH data and predicted gene-expression profiles, respectively. All genes from the ISH data are shown. **e** Violin plot for the predictive accuracies of Perler for the LOOCV experiments for all genes in the reference ISH data. **f** Prediction of non-landmark genes. In addition to the monotonic gene-expression profiles, non-monotonic gene-expression profiles are observed (Hamp, Igfbp2, Cyp8b1, and Mup3)[7]. **g, h** Application of Perler for the mouse visual cortex. **g** Cross-validation (CV) study (tenfold). The upper and lower panels show the referenced ISH data and predicted gene-expression profiles, respectively. **h** ROC curve for the CV experiments of genes shown in **g**. **i** Violin plot representing the predictive accuracies of Perler for the tenfold CV experiments of all genes in the reference ISH dataset according to ROC score. The median ROC score is 0.59. Source data are provided in a Source data file.

It is worth mentioning a recent method called novoSpaRc[32]. This method proposed a new concept for predicting spatial expression patterns using the physical information of cells in tissue, which enables these predictions with little or no information regarding ISH gene-expression patterns. However, in practice, their predictive ability using *Drosophila* scRNA-seq data is unsatisfactory at single-cell resolution; therefore, this concept of using cellular information remains challenging. As a focus of

future study, it would be interesting to extend our generative model to introduce prior knowledge of physical information.

We demonstrated that Perler can integrate two distinct datasets of RNA-expression profiles, while also avoiding overfitting to the reference. These features suggest that Perler could be a suitable theoretical framework for integrating not only two RNA-expression datasets, but also two single-cell datasets with different modalities, such as chromatin accessibility measured by a single-

cell assay for transposase-accessible chromatin, using sequencing and DNA methylation measured by chromatin immunoprecipitation sequencing. Particularly in terms of multi-omics analysis, where datasets from two different modalities do not exactly match and are often sampled from different individuals and using different time intervals[33,34], Perler can potentially help integrate different types of single-cell genomics data. Thus, Perler provides a powerful and generalized framework for revealing the heterogeneity of multicellular systems.

## Methods

We developed a method to reconstruct spatial gene-expression profiles from an scRNA-seq dataset via comparison with a spatial reference map measured by ISH-based methods. In the spatial reference map, landmark gene-expression vectors ($D$ genes; e.g., $D = 84$ in early *D. melanogaster* embryos) are available for all cells, whose locations in the tissue are known. The landmark gene-expression vector of cell $k$ is represented as $\boldsymbol{h_k} = (h_{k,1}, h_{k,2}, ..., h_{k,D})^{\mathrm{T}}$, where cells are indexed by $k$ ($k \in \{1,2,..., K\}$), and $K$ is the total number of cells in the tissue of interest. By contrast, in an scRNA-seq dataset, genome-wide expression ($D'$ genes; e.g., $D' = 8924$ in early *D. melanogaster* embryos) lack information regarding cell location in tissue. The genome-wide expression vector of cell $n$ is represented as $\boldsymbol{y_n} = (y_{n,1}, y_{n,2}, ..., y_{n,D'})^{\mathrm{T}}$, where cells are indexed by $n$ ($n \in \{1,2,..., N\}$), and $N$ is the total number of cells used for scRNA-seq measurement.

**Observation model**. We modeled the difference between scRNA-seq and ISH measurements as

$$y_i = a_i h_i + b_i + c_i \xi_i \qquad (1)$$

where $y_i$ and $h_i$ indicate expression levels of landmark gene $i$ measured by scRNA-seq and ISH experiments, respectively; $\xi_i$ indicates Gaussian noise with zero mean and unit variance; and $a_i$, $b_i$, and $c_i$ are constant parameters for gene $i$, which are interpreted as scale difference amplification rates, background signals, and noise intensities, respectively.

We reduced the dimensionality of the genes to change Eq. (1) to

$$x_j = a_j r_j + b_j + c_j \xi_j \qquad (2)$$

where $x_j$ and $r_j$ indicate expression levels of metagene $j$ for scRNA-seq and ISH in the lower dimensional space, $j \in \{1, 2, ..., M\}$; and $M$ indicates the number of metagenes. In vector–matrix representation, the Eq. (2) is written as:

$$\mathbf{x} = \mathbf{Ar} + \mathbf{b} + \mathbf{C\xi} \qquad (3)$$

where $\mathbf{x} = (x_1, x_2, ..., x_M)^{\mathrm{T}}$, $\mathbf{r} = (r_1, r_2, ..., r_M)^{\mathrm{T}}$, $\mathbf{A} = \mathrm{diag}(a_1, a_2, ..., a_M)$, $\mathbf{b} = (b_1, b_2, ..., b_M)^{\mathrm{T}}$, $\mathbf{C} = \mathrm{diag}(c_1, c_2, ..., c_M)$, and $\mathbf{\xi} = (\xi_1, \xi_2, ..., \xi_M)^{\mathrm{T}}$.

**Metagene representation in lower dimensional space**. The dimensionalities of both scRNA-seq and reference data were reduced by PLSC analysis[19]. PLSC can extract the correlated coordinates from both datasets. In PLSC analysis, the cross-correlation matrix of scRNA-seq and ISH data is first calculated as:

$$\mathbf{W} = \mathbf{Y}^{\mathrm{T}} \mathbf{H} \qquad (4)$$

where $\mathbf{Y}$ and $\mathbf{H}$ indicate a $D \times N$ scRNA-seq data matrix with $D$ landmark genes and $N$ cells, and a $D \times K$ ISH data matrix with $D$ landmark genes and $K$ cells, respectively. $\mathbf{W}$ is then subjected to singular value decomposition as:

$$\mathbf{W} \simeq \mathbf{U}^{\mathrm{T}} \mathbf{\Delta V} \qquad (5)$$

where $\mathbf{U}$, $\mathbf{\Delta}$, and $\mathbf{V}$ indicate the $M \times N$ singular vector matrices, the $M \times M$ diagonal matrix, and $M \times K$ singular vector matrices, respectively, with $M$ representing the reduced dimension (i.e., the number of metagenes). In this study, the metagene vectors for scRNA-seq ($\mathbf{x}_n$) and the reference data ($\mathbf{r}_k$) were, respectively, calculated by:

$$\mathbf{x}_n = \mathbf{\Delta u}_n \qquad (6)$$

$$\mathbf{r}_k = \mathbf{\Delta v}_k \qquad (7)$$

where $\mathbf{u}_n$ and $\mathbf{v}_k$ indicate the $n$th row vector of $\mathbf{U}$ and the $k$th row vector of $\mathbf{V}$, respectively.

**A Gaussian mixture model (GMM) for scRNA-seq observation**. We used Eq. (3) to transform ISH observations into scRNA-seq observations. To infer from which cells in the tissue the scRNA-seq observations originated, we developed a generative model for metagene-expression vectors for scRNA-seq data $\mathbf{x}$, which was expressed by a $K$-components GMM:

$$P(\mathbf{x}|\boldsymbol{\theta}) = \sum_{k=1}^{K} \pi_k N(\mathbf{x}|\boldsymbol{\mu}_k, \boldsymbol{\Sigma}), \qquad (8)$$

where

$$\boldsymbol{\mu}_k = \mathbf{Ar}_k + \mathbf{b}, \qquad (9)$$

$\boldsymbol{\Sigma} = \mathrm{diag}(\sigma_1^2, \sigma_2^2, ..., \sigma_M^2)$ ($\sigma_j = c_j$), $N(\mathbf{x}|\boldsymbol{\mu}, \boldsymbol{\Sigma})$ indicates a multivariate Gaussian distribution with mean and variance-covariance matrix $\boldsymbol{\Sigma}$, and $\pi_k$ is the probability that $\mathbf{x}$ originated from cell $k$ in the tissue. Note that $\mathbf{A}$, $\mathbf{b}$, and $\boldsymbol{\Sigma}$ are unknown parameters that need to be estimated.

The log of likelihood function of this GMM model is given by:

$$L(\boldsymbol{\theta}) = \sum_{n=1}^{N} \ln \left( \sum_{k=1}^{K} \pi_k N(\mathbf{x}_n|\boldsymbol{\mu}_k, \boldsymbol{\Sigma}) \right) \qquad (10)$$

where $\boldsymbol{\theta}$ indicates a set of the parameters $\boldsymbol{\theta} \in \{\boldsymbol{\pi}, \mathbf{A}, \mathbf{b}, \boldsymbol{\Sigma}\}$ and $\boldsymbol{\pi} = (\pi_1, \pi_2, ..., \pi_M)^{\mathrm{T}}$.

**EM algorithm (the first step in Perler)**. To estimate the unknown parameters ($\boldsymbol{\pi}$, $\mathbf{A}$, $\mathbf{b}$, and $\boldsymbol{\Sigma}$), we maximize the log likelihood function using the EM algorithm. In the E step, based on the current parameter values, we calculated the responsibility, which represents the posterior probability that scRNA-seq vector $\mathbf{x}_n$ was derived from cell $k$ in the tissue as:

$$\gamma_{nk} = \frac{\pi_k N\left(\mathbf{x}_n|\mathbf{A}^{(\mathrm{old})}\mathbf{r}_k + \mathbf{b}^{(\mathrm{old})}, \boldsymbol{\Sigma}^{(\mathrm{old})}\right)}{\sum_j^K \pi_j N\left(\mathbf{x}_n|\mathbf{A}^{(\mathrm{old})}\mathbf{r}_k + \mathbf{b}^{(\mathrm{old})}, \boldsymbol{\Sigma}^{(\mathrm{old})}\right)} \qquad (11)$$

In the M step, we optimize the parameter values in order to maximize the log likelihood function based on the current responsibilities. These parameter values are updated as follows:

$$\pi_k^{(\mathrm{new})} = \frac{\sum_j^K \gamma_{nk}}{N}, \qquad (12)$$

$$a_i^{(\mathrm{new})} = \frac{\psi_i - \chi_i b_i^{(\mathrm{new})}}{\omega_i}, \qquad (13)$$

$$b_i^{(\mathrm{new})} = \frac{\omega_i \phi_i - \psi_i \chi_i}{N \omega_i - \chi_i^2}, \qquad (14)$$

$$\sigma_i^{2(\mathrm{new})} = \frac{1}{N} \sum_{n=1}^{N} \sum_{k=1}^{K} \gamma_{nk} \left(x_{ni} - a_i^{(\mathrm{new})} r_{ki} - b_i^{(\mathrm{new})}\right)^2, \qquad (15)$$

where

$$\phi_i = \sum_{n=1}^{N} \sum_{k=1}^{K} \gamma_{nk} x_{ni}, \qquad (16)$$

$$\chi_i = \sum_{n=1}^{N} \sum_{k=1}^{K} \gamma_{nk} r_{ni}, \qquad (17)$$

$$\psi_i = \sum_{n=1}^{N} \sum_{k=1}^{K} \gamma_{nk} x_{ni} r_{ni}, \qquad (18)$$

$$\omega_i = \sum_{n=1}^{N} \sum_{k=1}^{K} \gamma_{nk} r_{ni}^2. \qquad (19)$$

The detailed derivation for these equations is presented in a later subsection. The E and M steps iterate until the log likelihood function converges, after which the obtained estimated parameters $\hat{\boldsymbol{\theta}} \in \{\hat{\boldsymbol{\pi}}, \hat{\mathbf{A}}, \hat{\mathbf{b}}, \hat{\boldsymbol{\Sigma}}\}$. $\hat{\boldsymbol{\mu}}_k$ are given as:

$$\hat{\boldsymbol{\mu}}_k = \hat{\mathbf{A}}\mathbf{r}_k + \hat{\mathbf{b}}, \qquad (20)$$

describing the mapped metagene-expression vector of cell $k$ measured by ISH. Note that $\hat{\boldsymbol{\mu}}_k$ is the metagene-expression vector in the scRNA-seq space.

In Perler, updating $\pi_k$ is optional. Ideally, $\pi_k$ should be proportional to the number of cells within region $k$. In our study, $\pi_k$ was fixed as $1/K$ for *Drosophila*, zebrafish, and mouse cortex data as the tissue was equally divided in the ISH data, whereas $\pi_k$ values were fixed to the area of each zone in the mammalian liver data. Note, fixing $\pi_k$ accelerated the convergence of the EM algorithm compared with optimizing $\pi_k$ (Supplementary Figs. 25 and 26).

For the initialization of parameter values, we selected the values of $a_i$ and $b_i$ such that mean and variance of each element of $x_{ni}$ and $r_{ki}$ were the same and selected the $c_i$ values as standard deviation of $x_{ni}$.

**Spatial reconstruction (the second step in Perler)**. We reconstructed/predicted the gene-expression vector by weighted averaging all scRNA-seq data points as

$$\bar{\mathbf{y}}_k = \sum_{n=1}^{N} \frac{w_{nk} \mathbf{y}_n}{\sum_{j=1}^{N} w_{jk}}, \qquad (21)$$

where $y_n$ indicates the $n$th scRNA-seq data point ($D$-component vector). $w_{nk}$ is calculated by

$$w_{nk} = \frac{\pi_k \exp\left(-\alpha D_{nk}^2 - \beta D_{nk} - \delta\right)}{\sum_{j=1}^{K} \pi_j \exp\left(-\alpha D_{nj}^2 - \beta D_{nj} - \delta\right)}, \qquad (22)$$

where $\alpha$, $\beta$, and $\delta$ are positive constants. Note that $\delta$ in the numerator and denominator of Eq. (22) are canceled out. $D_{nk}$ indicates Mahalanobis distance between scRNA-seq data point $\mathbf{x}_n$ and cell $k$:

$$D_{nk} = \sqrt{(\mathbf{x}_n - \hat{\boldsymbol{\mu}}_k)^{\mathrm{T}} \hat{\boldsymbol{\Sigma}}^{-1} (\mathbf{x}_n - \hat{\boldsymbol{\mu}}_k)}. \tag{23}$$

If $\alpha = 1/2$ and $\beta = 0$, $w_{nk}$ is exactly the posterior probability that scRNA-seq data point $\mathbf{x}_n$ is generated by cell $k$. Note that Eq. (21) has a similar structure to the Nadaraya–Watson model[14]. Values of $\alpha$ and $\beta$ are determined by CV.

**Weight sensitivity to the small perturbation of metagene**. We calculated differentiation of weight with respect to each metagene-expression level of scRNA-seq data point as:

$$\frac{dw_{nk}}{dx_{ni}} = \frac{w_{nk}(1 - w_{nk})(-2\alpha D_{nk} - \beta)}{D_{nk}} \frac{x_{ni} - \mu_{ki}}{\sigma_i^2} \tag{24}$$

$$\propto \frac{x_{ni} - \mu_{ki}}{\sigma_i^2} \tag{25}$$

These equations show that as $x_{ni}$ moves away from $\mu_{ki}$, weights decrease with a rate inversely proportional to the estimated noise of metagene $i$. This relationship indicates that small changes in unreliable genes with high noise levels has little effect on the weight, while small changes in reliable genes with low levels of noise have a large effect on weight. Therefore, Perler can reconstruct gene-expression profiles in a noise-resistant manner by accounting for the reliability of each gene through weight determination.

**Hyperparameter optimization**. We optimized the hyperparameters $\alpha$ and $\beta$ of the weighting function by LOOCV, in order to fit the predicted gene expression to the referenced gene expression measured by ISH. To this end, we removed one of the landmark genes from the ISH data and used this dataset to predict the spatial gene-expression profile of the removed landmark gene with the fixed hyperparameters in Perler. This LOO prediction was repeated for every landmark gene. We then quantitatively evaluated the predictive performance of these hyperparameters according to the mutual information existing between the predicted expression and referenced expression of all landmark genes:

$$J = -\frac{1}{2} \sum_{i}^{D} \ln\{1 - \rho_i(\alpha, \beta)^2\}, \tag{26}$$

where $J$ is the approximated mutual information between the predicted and referenced gene expression. $\rho_i(\alpha, \beta)$ indicates the Pearson's correlation coefficient between the predicted spatial expression pattern of each landmark gene $i$ and its reference ISH data as:

$$\rho_i(\alpha, \beta) = \frac{\sum_k^K (\bar{y}_{ki} - \langle \bar{y}_i \rangle)(h_{ki} - \langle h_i \rangle)}{\sqrt{\sum_k^K (\bar{y}_{ki} - \langle \bar{y}_i \rangle)^2} \sqrt{\sum_k^K (h_{ki} - \langle h_i \rangle)^2}}, \tag{27}$$

where

$$\langle \bar{y}_i \rangle = \frac{1}{K} \sum_{k=1}^{K} \bar{y}_{ki}, \text{ and} \tag{28}$$

$$\langle h_i \rangle = \frac{1}{K} \sum_{k=1}^{K} h_{ki}. \tag{29}$$

The derivation of $J$ is described in a later subsection. Here, we optimized $\alpha$ and $\beta$ by grid search in order to maximize the mutual information, $J$. We then used the optimized hyperparameters to predict the spatial profile of non-landmark genes (Fig. 3 and Supplementary Fig. 5). To evaluate the predictive performance of Perler (Fig. 3), we removed each landmark gene from the mutual information and re-optimized the hyperparameters. This re-optimization is repeated for every landmark gene. Note that for the zebrafish embryo data, we used the ROC score instead of the correlation coefficient, because only the binary ISH data was available. In addition, for the mouse visual cortex data, we conducted tenfold CV because of the massive computational cost of LOOCV for the large number of landmark genes (1020 genes).

**Data acquisition and preprocessing**. For *D. melanogaster* reconstruction, we used scRNA-seq and ISH data at *Drosophila* Virtual Expression eXplorer (DVEX; https://shiny.mdc-berlin.de/DVEX/[5]), which was originally used for DistMap[5]. In these datasets, the number of scRNA-seq data points is 1297, whereas the number of cells to be estimated in the embryos is 3039. The expressed mRNA counts in this scRNA-seq dataset were already log normalized according to the total number of unique molecular identifiers for each cell. For each gene, we subtracted the average expression from the scRNA-seq data. In addition, the ISH data were log-scaled and subtracted average expression from this ISH data, as same as the scRNA-seq data.

For reconstruction of the early zebrafish embryos, we acquired the public scRNA-seq and ISH data from the Satija Lab homepage (https://satijalab.org/[4]), with these data originally used by Seurat (v.1)[4]. In these data, the number of scRNA-seq data points is 851, whereas the number of subregions to be estimated in the embryos is 64. Note that the ISH data were binary. Similar to the *Drosophila*

data, we log-scaled both scRNA-seq and ISH datasets and subtracted the average expression of each gene.

For reconstruction of the mammalian liver, we used scRNA-seq and smFISH data provided by Halpern et al.[7]. In these data, the number of scRNA-seq data points is 1415, whereas the number of zones to be estimated in the embryos is 9. Because multiple samples were provided in the smFISH data, we calculated their average at each tissue location for Perler, followed by log-scaling both the scRNA-seq and smFISH data and subtracting the average expression of each gene.

For reconstruction of the mouse visual cortex, we used scRNA-seq data provided by the Allen Brain Institute[17] and Drop-seq data provided by Saunders et al.[31]. For ISH data, we used smFISH data provided by Wang et al.[16], respectively, which were originally used for Seurat (v.3)[9]. The number of scRNA-seq data points is 14,739 and 194,027, respectively, whereas the number of cells to be estimated in the cortex is 1549. We log-scaled both the scRNA-seq and smFISH data, and subtracted the average expression of each gene.

**Data analysis**. We used the newly developed method, Perler for data analysis. Perler was deposited at GitHub (see "Code availability"). Perler is built on Python 3.8.3.

All other software used in this study is publicly available: Numpy==1.19.5 (https://numpy.org/) for calculation; Scipy==1.6.1 (https://www.scipy.org/) for calculation; joblib==1.0.0 (https://joblib.readthedocs.io/en/latest/index.html) for calculation; scikit-learn==0.24.1 (https://scikit-learn.org/) for traditional machine learning (e.g., PCA and NN method); Matplotlib==3.3.3 (https://matplotlib.org/) for data visualization; Seaborn==0.11.1 (https://seaborn.pydata.org/) for data visualization; and pandas==1.2.1 (https://pandas.pydata.org/) for reading data frames. For the visualization of zebrafish embryos, we used "zf.insitu.vec.lateral" function of Seurat (v. ≥ 1.2).

**Data visualization**. For *D. melanogaster*, we visualized the reconstructed gene-expression profile at single-cell resolution by using the three-dimensional coordinates of all cells from DVEX (https://shiny.mdc-berlin.de/DVEX/[5]). Because the embryo is bilaterally symmetric, we mapped the reconstructed spatial gene-expression levels of the 3039 cells in the right-half embryo. According to the previous study[5], we then mirrored the spatial gene-expression levels of the right-half cells to the remaining cells in left-half embryo. In the case of the early zebrafish embryos, we visualized the reconstructed gene expression using the "zf.insitu.vec. lateral" function of Seurat (v. ≥ 1.2)[4]. In the case of the mammalian liver, we visualized the reconstructed gene expression as a heatmap. In the case of the mouse visual cortex, we visualized the reconstructed gene expression at single-cell resolution. We used two-dimensional coordinates of all cells within cortical slices provided by Wang et al.[16].

**Derivation of the EM algorithm**. The goal of the EM algorithm is to maximize the likelihood function $p(\mathbf{X} | \boldsymbol{\theta})$ with respect to $\boldsymbol{\theta}$, where $\mathbf{X} = \{\mathbf{x}_1, \mathbf{x}_2, \ldots, \mathbf{x}_N\}$ and $\boldsymbol{\theta} = \{\boldsymbol{\pi}, \mathbf{A}, \mathbf{b}, \boldsymbol{\Sigma}\}$. The generative model of scRNA-seq data point $\mathbf{x}$ with latent variables $\mathbf{z}$ is formulated, as follows. The probability distribution of $\mathbf{z}$ is:

$$P(\mathbf{z}) = \prod_{k=1}^{K} \pi_k^{z_k}, \tag{30}$$

where $\mathbf{z}$ is a vector in a one-of-$K$ representation that shows from which cells/ regions in tissue a scRNA-seq sample originated; $z_k$ is the $k$th element of $\mathbf{z}$; $K$ is the number of the elements of the latent variables $\mathbf{z}$ equal to the number of cells in the tissue; and $\pi_k$ is probability that $z_k = 1$. The probability distribution of $\mathbf{x}$ conditioned by $\mathbf{z}$ is:

$$P(\mathbf{x}|\mathbf{z}) = \prod_{k=1}^{K} N(\mathbf{x}|\mathbf{A}\mathbf{r}_k + \mathbf{b}_k, \boldsymbol{\Sigma})^{z_k}, \tag{31}$$

where $N(\mathbf{x}|\boldsymbol{\mu}, \boldsymbol{\Sigma})$ indicates a Gaussian distribution with mean $\boldsymbol{\mu}$ and variance $\boldsymbol{\Sigma}$; $\mathbf{A}_k$ is the $M \times M$ diagonal matrix; $\mathbf{b}_k$ indicates the $M$ elements vector in Eq. (3); and $\mathbf{r}_k$ indicates the $M$ elements vector describing the metagene-expression level in cell $k$. The joint probability distribution of $\mathbf{x}$ and $\mathbf{z}$ is:

$$P(\mathbf{x},\mathbf{z}) = \prod_{k=1}^{K} \{\pi_k N(\mathbf{x}|\mathbf{A}\mathbf{r}_k + \mathbf{b}_k, \boldsymbol{\Sigma})\}^{z_k}. \tag{32}$$

Note that the marginalized distribution of $\mathbf{z}$ becomes Eq. (8). The likelihood function for the complete dataset $\{\mathbf{X}, \mathbf{Z}\}$ is given as:

$$P(\mathbf{X},\mathbf{Z}) = \prod_{n=1}^{N} \prod_{k=1}^{K} \{\pi_k N(\mathbf{x}_n|\mathbf{A}\mathbf{r}_k + \mathbf{b}, \boldsymbol{\Sigma})\}^{z_{nk}}, \tag{33}$$

where $\mathbf{Z} = \{\mathbf{z}_1, \mathbf{z}_2, \ldots, \mathbf{z}_N\}$. Therefore, the expectation of its log likelihood function over the posterior distribution of $P(\mathbf{Z}|\mathbf{X}, \boldsymbol{\theta}^{(\mathrm{old})})$ becomes:

$$Q(\boldsymbol{\theta}, \boldsymbol{\theta}^{(\mathrm{old})}) = \sum_{\mathbf{Z}} P(\mathbf{Z}|\mathbf{X}, \boldsymbol{\theta}^{(\mathrm{old})}) \ln P(\mathbf{X}, \mathbf{Z}|\boldsymbol{\theta}) \tag{34}$$

$$= \sum_{n=1}^{N} \sum_{k=1}^{K} \gamma_{nk} \ln \pi_k + \gamma_{nk} \ln N(\mathbf{x}_n|\mathbf{A}\mathbf{r}_k + \mathbf{b}, \boldsymbol{\Sigma}), \tag{35}$$

where $\gamma_{nk}$ is the expectation of $z_{nk}$ over $P(\mathbf{Z}|\mathbf{X}, \boldsymbol{\theta}^{(\text{old})})$ given as:

$$\gamma_{nk} = \sum_{\mathbf{z}_n} z_{nk} P\left(\mathbf{z}_n | \mathbf{x}_n, \boldsymbol{\theta}^{(\text{old})}\right). \tag{36}$$

According to Bayes' theorem:

$$P\left(\mathbf{z}_n | \mathbf{x}_n, \boldsymbol{\theta}^{(\text{old})}\right) = \frac{P\left(\mathbf{x}_n | \mathbf{z}_n, \boldsymbol{\theta}^{(\text{old})}\right) P\left(\mathbf{z}_n, \boldsymbol{\theta}^{(\text{old})}\right)}{\sum_{\mathbf{z}_n} P\left(\mathbf{x}_n | \mathbf{z}_n, \boldsymbol{\theta}^{(\text{old})}\right) P\left(\mathbf{z}_n, \boldsymbol{\theta}^{(\text{old})}\right)}, \tag{37}$$

where $P(z_{nk} = 1|\mathbf{x}_n)$ becomes Eq. (11).

In the E step, $\gamma_{nk}$ is calculated based on the current parameter values of $\boldsymbol{\theta}^{(\text{old})}$. In the M step, we update the parameter values $\boldsymbol{\theta}$ by maximizing the Q-function as:

$$\boldsymbol{\theta}^{(\text{new})} = \underset{\boldsymbol{\theta}}{\arg\max} \, Q\left(\boldsymbol{\theta}, \boldsymbol{\theta}^{(\text{old})}\right). \tag{38}$$

The maximization of $Q(\boldsymbol{\theta}, \boldsymbol{\theta}^{(\text{old})})$ with respective to $\mathbf{A}$, $\mathbf{b}$, and $\boldsymbol{\Sigma}$ is achieved by $\partial Q/\partial \mathbf{A} = 0$, $\partial Q/\partial \mathbf{b} = 0$, and $\partial Q/\partial \boldsymbol{\Sigma} = 0$, leading to Eqs. (13–19). $\boldsymbol{\pi}$ is updated by introducing a Lagrange multiplier to enforce the constraint $\sum_{k=1}^{K} \pi_k = 1$, leading to Eq. (12).

**Derivation of mutual information**. We derived Eq. (26) by approximating the following mutual information between the reconstructed spatial expression pattern of the landmark genes and their reference map:

$$\mathrm{I}(\bar{\mathbf{y}}, \mathbf{h}) = \int \int P(\bar{\mathbf{y}}, \mathbf{h}) \ln \frac{P(\bar{\mathbf{y}}, \mathbf{h})}{P(\bar{\mathbf{y}}) P(\mathbf{h})} \mathrm{d}\bar{\mathbf{y}} \mathrm{d}\mathbf{h}, \tag{39}$$

where $\bar{\mathbf{y}} = (\bar{y}_1, \bar{y}_2, \ldots, \bar{y}_D)$, $\mathbf{h} = (h_1, h_2, \ldots, h_D)$, and $\bar{y}_i$ and $h_i$ indicate random variables representing the predicted and referenced expression levels of landmark gene $i$, respectively, and $P(\bar{\mathbf{y}}, \mathbf{h})$ indicates the joint probability distribution of $\bar{\mathbf{y}}$ and $\mathbf{h}$. Here, we assumed that spatial expressions of landmark genes are independent from one another, which leads to:

$$\mathrm{I}(\bar{\mathbf{y}}, \mathbf{h}) = \sum_i^D \int \int P(\bar{y}_i, h_i) \ln \frac{P(\bar{y}_i, h_i)}{P(\bar{y}_i) P(h_i)} \mathrm{d}\bar{y}_i \mathrm{d}h_i. \tag{40}$$

We calculated $\mathrm{I}(\bar{\mathbf{y}}, \mathbf{h})$ by assuming $P(\bar{y}_i, h_i)$ as a bivariate Gaussian distribution and obtained:

$$\mathrm{I}(\bar{\mathbf{y}}, \mathbf{h}) = -\frac{1}{2} \sum_i^D \ln \left( 1 - \rho_i(\alpha, \beta)^2 \right), \tag{41}$$

where $\rho_i(\alpha, \beta)$ denotes the calculated Pearson's correlation coefficient calculated.

**Reporting summary**. Further information on research design is available in the Nature Research Reporting Summary linked to this article.

## Data availability

This study constituted a reanalysis of existing data. The detailed data are available at the following sites: *Drosophila* embryo datasets from *Drosophila* Virtual Expression eXplorer (DVEX); Zebrafish embryo datasets from Satija Lab homepage (https://satijalab.org/); mammalian liver datasets from Gene Expression Omnibus (GEO) with the accession code GSE84498; STARmap data from STARmap Resources website (https://www.starmapresources.com/; mouse cortex); SMART-seq2 data from Cell Types Database website of the Allen Institute for Brain Science (http://celltypes.brain-map.org/api/v2/well_known_file_download/694413985; mouse cortex); and Drop-seq data from Drop-viz website (http://dropviz.org/; mouse cortex). Source data are provided with this paper.

## Code availability

Perler is developed under python 3.8 on GitHub (https://github.com/yasokochi/Perler)[35]. The minimal usage of Perler is provided in Supplementary Table 6, and the selected parameters in the manuscript are provided in Supplementary Table 7. The running time and the memory usages on a MacBook Pro (2.3 GHz 8-Core Intel Core i9, 64GB) are also provided in Supplementary Table 8.

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

## Acknowledgements

This study was supported in part by the Cooperative Study Program of Exploratory Research Center on Life and Living Systems (ExCELLS; program Nos. 18-201, 19-102, and 19-202 to H.N.); Moonshot R&D–MILLENNIA Program (Grant No.: JPMJMS2024-9) by JST; a Grant-in-Aid for Young Scientists (B) (19H04776 and 21H03541 to H.N.), a Grant-in-Aid for Scientific Research (B) (17KT0021 to T.K.), and a JSPS research fellowship for young scientist (to S.S.) from the Japan Society for the Promotion of Science (JSPS); the Naito Foundation (to T.K.); and the Keihanshin Consortium for Fostering the Next Generation of Global Leaders in

Research (K-CONNEX) established by the program of Building of Consortia for the Development of Human Resources in Science and Technology, MEXT (to T.K.).

## Author contributions

H.N., S.S., and T.K. conceived the project. Y.O., H.N., and K.N. developed the method, Y.O. implemented the software, and Y.O. and S.S. analyzed data. Y.O. and H.N. wrote the manuscript with input from all authors.

## Competing interests

The authors declare no competing interests.
