## [Peer Review File · Nature Communications]

REVIEWER COMMENTS

Reviewer #1 (Remarks to the Author):

The authors proposed a new computational method to predict spatial gene expression based on ISH data and scRNA-seq data. While the question is pivotal to various biological questions and the proposed methods seems to have excellent performance compared with the existing methods, the current manuscript should be substantially revised to clearly and logically demonstrate the robustness and effectiveness of the method.

1. It is impressive to show various beautiful figures of gene spatial expression. However, such figures are not informative or convictive. Quantitative metrics (such as in Figure 2d) should be used to show the validity, robustness and efficacy of the new algorithm.
2. Specially, the authors proposed a linear mapping method to establish the relationships of landmark genes between ISH and scRNA-seq. However, readers cannot find data supporting the effectiveness of the linear mapping method, in particular on the gene levels. Figure 2b is not convincing as it may be over-stretched.
3. Dimension reduction is used before starting the linear mapping and spatial reconstruction. However, the validity and role of dimension reduction in the whole prediction is not shown. In particular, in figure 1, it is weird to derive scRNA-seq metagenes from ISH data and ISH metagenes from scRNA-seq. The authors should go back to the gene levels to check to what extent the dimension reduction is properly performed.
4. One important concern is the resolution of the prediction. Given ISH and scRNA-seq data, the spatial origins of single cells can be inferred by the authors' method. The authors should evaluate how those single cells were spatially resolved by calculating the resolution. Although the authors showed such data in Figure 1d, it is not sufficient. In particular, the authors should show how the resolution was changed when the number of types of landmark genes changed on multiple independent datasets. Comparisons with other methods regarding this type of performance are also needed.
5. Throughout the manuscript, only one dataset is extensively used to show the performance of the method. Although the authors also evaluated on other datasets, the performance is less presented. Strict and comprehensive evaluation by quantitative metrics on multiple datasets are expected. One dataset is not convincing.
6. Parameter tuning and the computational cost should also be clearly presented for each dataset. Otherwise readers cannot know the generalization ability of the method.

Reviewer #3 (Remarks to the Author):

The authors present Perler, a probabilistic embryo reconstruction by linear evaluation of scRNAseq, although several methods have already been published using the same data sets, the authors claim that the mapping is done without using the underlying scRNA-seq data structure such as gene-expression noise. Authors claim also that given that methods are dependent upon nonlinear NN mapping, this leads to over-fitting to the reference ISH data. Furthermore, the authors nicely show that Perler can reconstruct a spatial gene-expression pattern and preserve the timing information of the scRNA-seq data. The article is well written and illustrated with good figures and although the accuracy of Perler compared to other methods seems better (.76 compared to ~0.6). The results regarding the precision of gene-expression profiles of the segment-polarity genes within each parasegment and the timing expression differences for pair-rule genes are a good proof of the accuracy of Perler. However, I am a bit surprised that the change in accuracy shown with Perler gives such different results and I think a lot more work needs to be done to give credibility to all results and conclusions (See below). Also I think that not-binarizing the scRNAseq and the generative model used are not innovative enough to support the publication of the manuscript.

Major comments:

The authors question overfitting of the data, but maybe should try a CV scheme such as the one described here, using the same data: <https://www.biorxiv.org/content/10.1101/796029v2> with the CV scheme to be obtained here: <https://github.com/dream-sctc/Data>

Also, in that manuscript a somehow similar approach is described in (see Hafemeister and also <https://www.synapse.org/#!Synapse:syn17061192/wiki/586352>)

Finally they claim higher accuracy than other methods, but they should compare using the metrics used in the DREAM challenge (<https://www.synapse.org/#!Synapse:syn15665609/wiki/582909>) using the CV scheme, see here: <https://github.com/dream-sctc/Scoring>

Minor comments:

-I am surprised by the loss of accuracy in all methods when doing LOOCV, this enhances the need to use as scheme as in the DREAM challenge for selecting less landmark genes and evaluating the algorithm's performance. LOOCV is really not satisfactory in this case.

Typos:
pg 5 EM algorithm

Draft Only

Responses to reviewers' comments

Reviewer 1

Remarks to the Author

The authors proposed a new computational method to predict spatial gene expression based on ISH data and scRNA-seq data. While the question is pivotal to various biological questions and the proposed methods seems to have excellent performance compared with the existing methods, the current manuscript should be substantially revised to clearly and logically demonstrate the robustness and effectiveness of the method.

We would like to appreciate the reviewer's constructive comments. We have improved our manuscript and prepared the response to each of the reviewer's comments and suggestions as shown below.

Comment 1

It is impressive to show various beautiful figures of gene spatial expression. However, such figures are not informative or convictive. Quantitative metrics (such as in Figure 2d) should be used to show the validity, robustness and efficacy of the new algorithm.

In this general comment, the reviewer strongly suggested us to more clearly show three characteristics (1. validity, 2. robustness and 3. efficacy) of our Perler algorithm. For each, we responded to the reviewer's specific comments below:

1. For the validity of Perler, we responded to the comments 2-3. In revision, we showed the validities of generative linear mapping, hyperparameter optimization, and dimensionality reduction.
2. For the robustness of Perler, we responded to the comments 4. In revision, we showed that Perler robustly maintained resolutions of the predictions of cell location and spatial gene expressions in the downsampled dataset.
3. For the efficacy of Perler, we responded in response to the comments 6. In revision, we quantified the computational efficacy of Perler. In addition, we significantly improved the efficacy by fixing one of the linear mapping parameters.

Comment 2

Specially, the authors proposed a linear mapping method to establish the relationships of landmark genes between ISH and scRNA-seq. However, readers cannot find data supporting the effectiveness of the linear mapping method, in particular on the gene levels. Figure 2b is not convincing as it may be over-stretched.

We agree with the reviewer's comment that we should show the validity of our linear mapping method in Perler. First, as shown in the previous manuscript for Drosophila data, we illustrated that the mapped ISH data points and scRNA-seq data points are consistently distributed in the PCA space (**Fig. 2a and Supplementary Fig. 2d**). In revision, as suggested by the reviewer, we also evaluated that on each metagene level, the expression levels in the mapped ISH data are more consistently distributed with those in scRNA-seq data than those in the unmapped ISH data (**Supplementary Fig. 3, lines 165-167**). These results clearly indicated

that our linear mapping in Perler correctly calibrated the difference between ISH and scRNA-seq data. We conducted this kind of analysis in other datasets for zebrafish, liver, and cortex (**Supplementary Fig. 2 and 3 [lines 163-167, and 168-170], 16 and 17 [lines 319-320], 18 and 19 [lines 325-326], 20 and 21 [lines 329-331], 22 and 23 [lines 333-335], and 25 and 26 [lines 493-497]**).

> Figure 2b is not convincing as it may be over-stretched.

Thank you for the suggestion for the role of hyperparameter optimization. Figure 2b showed that the hyperparameter optimization improved correlation between reference and predicted data for two genes (*ftz* and *gk*). But, these were not the selected cases. In revision, we additionally showed correlation improvement via the hyperparameter optimization for all landmark genes (**Supplementary Fig. 5, lines 174-176**). Additionally, we demonstrated that hyperparameter optimization significantly improved the whole performance of Perler in all datasets (Drosophila, zebrafish, liver, and cortex) (**Supplementary Fig. 11, lines 223-224, 319-320, 325-326, and 329-331**).

Comment 3

Dimension reduction is used before starting the linear mapping and spatial reconstruction. However, the validity and role of dimension reduction in the whole prediction is not shown. In particular, in figure 1, it is weird to derive scRNA-seq metagenes from ISH data and ISH metagenes from scRNA-seq. The authors should go back to the gene levels to check to what extent the dimension reduction is properly performed.

Thank you for an informative suggestion to check the role and validity of dimensionality reduction.

> In particular, in figure 1, it is weird to derive scRNA-seq metagenes from ISH data and ISH metagenes from scRNA-seq.

Sorry for confusion of our description of the dimensionality reduction (PLSC) caused by our mistake in previous figure 1. Correctly, scRNA-seq metagenes from scRNA-seq data and ISH metagenes from ISH data. In the current manuscript, we revised figure 1.

> The authors should go back to the gene levels to check to what extent the dimension reduction is properly performed.

Before responding to this comment, we confirmed that some genes redundantly exhibit similar expression patterns by calculating correlation coefficient between all pairs of the landmark genes in the revised manuscript (**Supplementary Fig. 1, lines 160-163**). It suggested that dimensionality reduction can be useful to reduce the redundancy of landmark genes.

To respond to this comment, we showed that the dimensionality reduction improved the predictive accuracy of Perler, compared with the case in the absence of the dimensionality reduction (**Supplementary Fig. 10, lines 220-223 and 319-320**). This kind of analysis was conducted in multiple data sets (Drosophila and zebrafish).

Comment 4

One important concern is the resolution of the prediction. Given ISH and scRNA-seq data, the origins of single cells can be inferred by the authors' method. The authors should evaluate how those single cells were spatially resolved by calculating the resolution. Although the authors showed such data in **Figure 1d***, it is not sufficient. In particular, the authors should show how the resolution was changed when the number of types of landmark genes changed on multiple independent datasets. Comparisons with other methods regarding this type of performance are also needed.

*** Note by authors: We guess Figure 1d is a typo for Figure 2d, because Figure 1d is not related to this comment.**

Thank you for this constructive suggestion about the robustness to the downsampled sets of landmark genes.

> **The authors should evaluate how those single cells were spatially resolved by calculating the resolution. Although the authors showed such data in Figure 1d, it is not sufficient. In particular, the authors should show how the resolution was changed when the number of types of landmark genes changed on multiple independent datasets.**

As suggested in this comment, we evaluated the spatial resolution of the inferred origins of the scRNA-seq data points. Previously, we calculated the **assigned specificity** only using the full landmark gene set (**Fig. 2d in previous version**). In revision, we performed **the same analysis** by using randomly downsampled landmark gene sets (**Supplementary Fig. 13, lines 229-230, 319-320, and 329-331**). As increasing the number of the landmark genes selected, we showed that the spatial resolution of the inference was saturated at the small set of landmark genes, suggesting that Perler was able to robustly infer the origin of scRNA-seq datapoint. Moreover, we showed that Perler was also able to robustly reconstruct the spatial gene expression patterns by using the randomly downsampled gene sets (**Supplementary Fig. 12, lines 225-227, 319-320, and 329-331**).

As suggested by reviewers in comment 5 below, we confirmed the robust performances of Perler by conducting the same analyses in other datasets (zebrafish and cortex). Taken together, we concluded that Perler has robustness to the downsampled datasets.

> **Comparisons with other methods regarding this type of performance are also needed.**

Thank you for bringing up this great point. In the previous manuscripts, we only compared Perler with other methods in terms of the reconstruction/prediction of the spatial gene expression patterns. As suggested here, we performed the comparison of Perler with the other methods (Liger, Seurat, and DistMap) in terms of the spatial resolution of the inferred origins of scRNAseq data points (**Supplementary Fig. 4, lines 171-173**), indicating that Perler's performance is superior to DistMap, but is equivalent to Liger and Seurat v.3.

Comment 5

Throughout the manuscript, only one dataset is extensively used to show the performance of the method. Although the authors also evaluated on other datasets, the performance is less presented. Strict and comprehensive evaluation by quantitative metrics on multiple datasets are expected. One dataset is not convincing.

Thank you for the suggestion. As described in our responses above, we extensively evaluated the performance of Perler on all datasets (Drosophila, zebrafish, liver, and cortex) (**Supplementary Fig. 3-5, 10-13, 17, 19, 21, 23, and 26**), as follows

- The effects of linear mapping on each metagene level in Perler were evaluated on all datasets in response to **Comment 2**.
- The effects of hyper-parameter optimization in Perler's performance were evaluated on all datasets in response to **Comment 2**.
- The effects of dimension reductions on Perler's performance were evaluated on all datasets in response to **Comment 3**.

In addition to these, we also showed the linear mapping of Perler properly integrated scRNA-seq and ISH data in all metagenes level (related to Fig. 2a, c, d and Sfig. 1) for all datasets, which was previously shown only in the Drosophila dataset (**Supplementary Fig. 2, 16, 18, 20, 22, and 25**).

Comment 6

Parameter tuning and the computational cost should also be clearly presented for each dataset. Otherwise readers cannot know the generalization ability of the method.

This is an important point for future users of Perler. In revision, we added the minimum usage of Perler (**Supplementary Table 6, lines 651-652**), listed the control parameters of Perler (**Supplementary Table 7, lines 652-653**), and summarized computational cost including running time and memory usage on our macbook pro (2.3 GHz 8-Core Intel Core i9, 64GB) (**Supplementary Table 8, lines 653-654**). In addition, we uploaded Perler source code as a python package on Github (<https://github.com/yasokochi/Perler>), in which the detailed tutorials of Perler usage for all datasets (Drosophila, zebrafish, liver, and cortex) were prepared.

Reviewer 3

Remarks to the Author

The authors present Perler, a probabilistic embryo reconstruction by linear evaluation of scRNAseq, although several methods have already been published using the same data sets, the authors claim that the mapping is done without using the underlying scRNA-seq data structure such as gene-expression noise. Authors claim also that given that methods are dependent upon nonlinear NN mapping, this leads to over-fitting to the reference ISH data. Furthermore, the authors nicely show that Perler can reconstruct a spatial gene-expression pattern and preserve the timing information of the scRNA-seq data. The article is well written and illustrated with good figures and although the accuracy of Perler compared to other methods seems better (.76 compared to ~0.6). The results regarding the precision of gene-expression profiles of the segment-polarity genes within each parasegment and the timing expression differences for pair-rule genes are a good proof of the accuracy of Perler.

We would like to appreciate the reviewer's invaluable comments. We have improved our manuscript and prepared the responses to the reviewer's comments and suggestions as shown below.

However, I am a bit surprised that the change in accuracy shown with Perler gives such different results and I think a lot more work needs to be done to give credibility to all results and conclusions (See below).

In revision, we did a lot more work to show the credibility of Perler. Please see our responses to the reviewer's major comments below.

Also I think that not-binarizing the scRNAseq and the generative model used are not innovative enough to support the publication of the manuscript.

We apologize for confusion caused by not explaining enough. We agree with the reviewer's comment that not-binarizing the scRNAseq and the generative model are not innovative.

We think that binarization of a scRNAseq data is unnatural and leads to loss of information. As mentioned in Introduction in the previous manuscript (**lines 59-61 in the current manuscript**), the previous methods, e.g, Liger and Seurat v3 addressed scRNAseq data as continuous variables. In addition, a generative model has been commonly used to represent data structure in the fields of statistics and machine learning, as used in the previous methods, e.g., Seurat. v1, Halpern et al. Therefore, we think that not-binarizing and a generative model themselves or their combination are not innovative. However, we want to emphasize there are three innovative points of Perler.

Firstly, in the first step of Perler, Perler can calibrate the difference between measurement properties of scRNA-seq and ISH in a biologically interpretable manner. To express the difference, we used a "linear mapping model" assuming that expression levels are linearly correlated between ISH and scRNA-seq measurements with gene-specific sensitivity, background signals, and noise intensity, as in equation (1). In contrast, there is no such calibration method based on the difference between measurement properties.

Secondly, in the second step of Perler, Perler can reliably reconstruct the gene expression pattern in a noise resistant manner. Specifically, Perler evaluated to what extent each gene is reliable for the reconstruction depending on the noise intensity (related to **Fig. 1d**) by using Mahalanobis' pairwise distances (equation (23), related to **Fig. 1c**). As a result, more reliable genes with low noises were more used for the reconstruction, whereas less reliable genes with high noises were less used. It should be stressed that such quantitative evaluation of gene reliability is possible only by a method using a generative model. On the other hand, existing methods (Liger, Seurat v.3, and DistMap) adopted distances independent of the noise intensity (e.g. Euclidean distance and MCC), because they were not based on generative models.

Lastly, the model-based linear mapping used in Perler has another benefit in terms of the performance of the reconstruction of gene expression patterns. Perler significantly improved reconstruction/prediction accuracy compared with other methods (Liger, Seurat v.3, and DistMap) (**Fig. 2e and f**). In a demonstration in *Drosophila* data, Perler successfully reconstructed 14-stripe patterns of the segment-polarity genes with a single-cell resolution (**Fig. 4**). Furthermore, Perler can preserve the timing information of scRNAseq data and robustly reconstruct the spatial gene expression patterns of the pair-rule genes, while this kind of robustness is not shown in other model-free methods (Liger, Seurat, and DistMap) (**Fig. 5**). We think that these results strengthen the importance of using model-based prediction of spatial gene expression patterns.

In the revised manuscript, we additionally mentioned these points in Discussion (**lines 366-393**).

Major comment 1

The authors question overfitting of the data, but maybe should try a CV scheme such as the one described here, using the same data: <https://www.biorxiv.org/content/10.1101/796029v2> with the CV scheme to be obtained here: <https://github.com/dream-sctc/Data>

Thank you for suggesting the CV scheme used in the DREAM Single-Cell Transcriptomics challenge (Tanevski et al., bioRxiv, 2019). This scheme was based on the scoring metrics:

- s1: how well the expression of the cell at the predicted location correlates to the expression from the reference atlas and included the variance of the predicted locations for each cell
- s2: the accuracy of the predicted location
- s3: how well the gene-wise spatial patterns were reconstructed

In revision, we performed the CV scheme (**Supplementary Table 2-4, lines 233-237**) in comparison with other methods (Liger and Seurat v.3). We found that Perler exhibited better performances than those of other methods (Liger and Seurat v3) for all scoring metrics (s1, s2, and s3).

Major comment 2

Also, in that manuscript a somehow similar approach is described in (see Hafemeister and also <https://www.synapse.org/#!Synapse:syn17061192/wiki/586352>)

Thank you for useful information about the top-ranked method in the Dream challenge by Dr. Hafemeister. In revision, we showed his method's scores as a reference (**Supplementary Table 2-4**).

Major comment 3

Finally they claim higher accuracy than other methods, but they should compare using the metrics used in the DREAM challenge (<https://www.synapse.org/#!Synapse:syn15665609/wiki/582909>) using the CV scheme, see here: <https://github.com/dream-sctc/Scoring>

As in our response to the major comment 1, we compared Perler with other methods (Liger and Seurat v3) by using the CV scheme.

Minor comment

I am surprised by the loss of accuracy in all methods when doing LOOCV, this enhances the need to use as scheme as in the DREAM challenge for selecting less landmark genes and evaluating the algorithm's performance. LOOCV is really not satisfactory in this case.

The reviewer concerned the loss of accuracy in LOOCV. In revision, we identified 11 landmark genes, which were poorly predicted in LOOCV. To clarify the difference between these 11 landmark genes and the other well-predicted genes, we examined correlated data structure among landmark genes. We found that these poorly-predicted genes have different expression patterns between ISH and scRNA-seq (**Supplementary fig. 9 and Supplementary Table 1, lines 204-213**), suggesting that the loss of accuracy was majorly caused by different correlated data structures between ISH and scRNA-seq possibly due to different measurement principles (hybridization vs. sequencing) and different biological conditions (developmental stages 5 vs. 6).

In addition, as suggested by the reviewer, we used the CV scheme in the DREAM challenge with selecting less landmark genes. We think that this suggestion is related to sub-challenges (SC1, SC2, and SC3) in the DREAM challenge, in which the CV scheme is performed with selected 20, 40, 60 landmark genes. In revision, we thus conducted the CV scheme in all sub-challenges (**Supplementary Table 2-4**).

Typo:

pg 5 EM algorithm

Thank you for pointing out our typo. In revision, we fixed it.

REVIEWERS' COMMENTS

Reviewer #1 (Remarks to the Author):

The authors have addressed my concerns and the manuscript is now acceptable for publication.

Reviewer #3 (Remarks to the Author):

We thank the authors for performing the suggested analysis regarding CV and comparison to other predictions. However it seems that they either did not understand the evaluation metrics or did not want to present the comparative results. It is clear that Perler does not fare better than the top DREAM algorithms (and in particular C. Hafemeister who used a method different from Seurat although having been part of the lab who developed that method).

All the sentences stating that: "Additionally, we showed that Perler can reconstruct a spatial gene-expression pattern that could not be fully predicted using previous methods, including Seurat (v.3), Liger, and DistMap." or "We also evaluated this performance of other methods Supplementary Fig. 4) and determined that this performance of Perler was superior to that of DistMap and equivalent to that of Liger and Seurat v.3." should be changed to something like "Perler was superior to that of DistMap and equivalent to that of Liger and Seurat v.3. but did not fare as well as the top methods developed in the DREAM challenge".

Also please add the correct reference for the DREAM paper: Life Science Alliance Sep 2020, 3 (11) e202000867;

It seems that the robustness claim of the method is valid specifically for Drosophila as the metagenes did not look as good for other organisms (see supp fig 21 and 23) this should be acknowledged.

Overall Perler is an original approach, but its limitations should be stated clearly.

REVIEWERS' COMMENTS

Reviewer #1 (Remarks to the Author):

The authors have addressed my concerns and the manuscript is now acceptable for publication.

Thank you for evaluating our revised manuscript.

Reviewer #3 (Remarks to the Author):

Comment 1

We thank the authors for performing the suggested analysis regarding CV and comparison to other predictions. However it seems that they either did not understand the evaluation metrics or did not want to present the comparative results. It is clear that Perler does not fare better than the top DREAM algorithms (and in particular C. Hafemeister who used a method different from Seurat although having been part of the lab who developed that method).

Firstly, the reviewer concerned our understanding of the evaluation metrics used in the DREAM challenge. In the revised manuscript, we rewrote the explanation of the metrics in the legends of **Supplementary Table 2-4** for the clarity as follows:

- s1: the correlation between the ISH expressions at the cells predicted by the proposed method and DistMap.
- s2: the inverse distance of the cells predicted by the proposed method to the most probable location predicted by DistMap.
- s3: the gene-wise correlations between the scRNA-seq expressions of landmark genes and the ISH expressions of the most probable cell predicted by the proposed method. Note that the calculated correlations are biasedly weighted by DistMap predictability for each gene.

Furthermore, we previously did not mention in the main text that these metrics were designed assuming that DistMap prediction was ground truth. In revision, we additionally mentioned this point in Results (**line 199-201**).

Secondly, the reviewers concerned the performance comparison between Perler and the top-ranked methods in the DREAM challenge. Although, in the previous manuscript, we showed the comparative results in the **Supplementary Table 2-4**, we agree with the reviewer's comment that we have to mention these results in the main text. In revision, we clearly mentioned that the top-ranked methods developed in the DREAM challenge had better performance than Perler in these metrics in Results (**line 197-198**).

Comment 2

All the sentences stating that: "Additionally, we showed that Perler can reconstruct a spatial gene-expression pattern that could not be fully predicted using previous methods, including Seurat (v.3), Liger, and DistMap." or "We also evaluated this performance of other methods Supplementary Fig. 4) and determined that this performance of Perler was superior to that of DistMap and equivalent to that of Liger and Seurat v.3." should be changed to something like "Perler was superior to that of DistMap and equivalent to that of Liger and Seurat v.3. but did not fare as well as the top methods developed in the DREAM challenge".

We agree with the reviewer's comment that the limitation of Perler should be clearly stated in terms of the DREAM challenge metrics. However, the metrics are really based on DistMap, which less performed both reconstruction and prediction than the other methods (Perler, Liger, and Seurat v.3; **Figure 2 and Supplementary Figure 7**). Thus, we considered that these metrics are not appropriate for evaluating the performance of the methods (Perler, Liger, and Seurat v.3). In revision, we mentioned this point at the subsection regarding the DREAM challenge in Results, as in our response to the comment 1 (line 197-198).

Comment 3

Also please add the correct reference for the DREAM paper: Life Science Alliance Sep 2020, 3 (11) e202000867;

Thank you for pointing out our mistake. We fixed it.

Comment 4

It seems that the robustness claim of the method is valid specifically for *Drosophila* as the metagenes did not look as good for other organisms (see supp fig 21 and 23) this should be acknowledged.

We think that the reviewer mistakenly understood the results of **Supplementary Figures 21 and 23**. The linear mapping properly calibrated the difference between ISH and scRNA-seq data on each metagene level for all the data of *Drosophila*, zebrafish, the mammalian liver, and the mouse cortex. Please compare "blue" and "black" lines before the linear mapping with "red" and "black" lines after the linear mapping.

Comment 5

Overall Perler is an original approach, but its limitations should be stated clearly.

As in our response to the comment 1-2, we mentioned this point in the line 197-198.

Draft Only